# TUFORMER: DATA-DRIVEN DESIGN OF TRANSFORMERS FOR IMPROVED GENERALIZATION OR EFFICIENCY

**Xiaoyu Liu**
Department of Computer Science
University of Maryland, College Park
MD 20740, USA
`xliu1231@umd.edu`

**Jiahao Su**
Department of Computer Science
University of Maryland, College Park
MD 20740, USA
`jiahaosu@umd.edu`

**Furong Huang**
Department of Computer Science
University of Maryland, College Park
MD 20740, USA
`furongh@umd.edu`

## ABSTRACT

Transformers are neural network architectures that achieve remarkable performance in many areas. However, the core component of Transformers, multi-head self-attention (MHSA), is mainly derived from heuristics, and the interactions across its components are not well understood. To address the problem, we first introduce a mathematically rigorous and yet intuitive tensor diagram representation of MHSA. Guided by tensor diagram representations, we propose a novel design, namely Tunable Transformers (Tuformers), by allowing data-driven weights across heads, whereas MHSA adopts pre-defined and fixed weights across heads, as will be explained in our paper. Tuformers naturally reveal a flexible design space that a user, depending on the needs, can choose a structure that has either improved performance (generalization error) or higher model efficiency. Any pre-trained Transformer can be an initialization of the corresponding Tuformer with trainable number of heads for efficient training and fine-tuning. Tuformers universally outperform Transformers on various tasks across multiple domains under a wide range of model sizes.[1]

## 1 INTRODUCTION

Transformer models are first introduced by Vaswani et al. (2017) in the context of sequence modelling. They have demonstrated impressive results on a wide variety of tasks in many fields, such as language model pre-training (Sanh et al., 2019), speech recognition (Moritz et al., 2020), image classification (Dosovitskiy et al., 2020), and generation (Parmar et al., 2018). The core component of Transformer models is the *multi-head self-attention* (MHSA) which is extended from the standard attention mechanism (Bahdanau et al., 2014). Each attention head in MHSA has a global receptive field, i.e., each token's representation is updated by attending to all other tokens, and $H$ attention heads are computed in parallel and concatenated together.

The current MHSA design is mainly derived from empirical studies or heuristics, leaving some unresolved challenges. **(1) Lack of solid theoretical understanding.** The theory behind MHSA is only starting to catch up with practice. The role of the components and their interactions in MHSA are not well understood rigorously from a theoretical perspective, which may lead to inefficiencies in the design. For example, Michel et al. (2019) show that most attention heads can be removed in the testing phase without much performance compromise, while Cordonnier et al. (2020) find empirical evidence of redundancy in key/query projections and propose a re-parameterization scheme. These works focus more on practical solutions, leading to questions about whether theoretical patterns exist in these designs. **(2) The number of heads is not trainable.** Intuitively, the heads in multi-heads

---

[1]code available at : https://github.com/umd-huang-lab/tuformer

are expected to capture different context information through each head. However, the number of heads is fixed in training. Thus, although we could tune the hyper-parameter, an exhaustive search would be time-consuming or not practical for large-scale tasks. **(3) Hard to analyze the expressive power.** Analyzing the expressive power (as will be defined in Definition 3) of a neural network, i.e., proving that some architectures are more expressive than others, is a non-trivial and challenging task. Such analysis has been done in Convolutional Neural Networks (CNNs) (LeCun et al., 1995; Cohen et al., 2016), Recurrent Neural Networks (RNNs) (Mikolov et al., 2010; Khrulkov et al., 2018), but no such work exists for interpreting MHSA or guiding the structural design.

In response to the above challenges, we first **interpret** MHSA from a tensor representation perspective using the intuitive graphical tool, the *tensor diagram*. Current prevalent descriptions of MHSA use flow charts to convey high-level intuitions, which could cause ambiguities. Therefore, it is inevitable to pair those flow charts with mathematical formulas to understand the mechanism precisely. However, these two separated descriptions create difficulties for interpretations and inspections of the operations implemented in MHSA. To address this issue, we propose a graphical representation of MHSA, which is both semantically intuitive and mathematically rigorous. Specifically, we modify and extend the vanilla *tensor diagrams* (Penrose, 1971), which conveniently allow for rigorous graphical representation of multi-linear operations between multi-dimensional arrays (i.e., higher-order tensors) to represent nonlinear operations.

Using tensor diagram representation, we project the current design of MHSA into its tensor form, which renders a holistic view of the weights in MHSA for better interpretation of the information flow and exchange among the components of MHSA. We then **propose a novel data-driven structure**, namely *Tunable-Head Self-Attention (THSA)*, which is a re-parameterization of the weight matrices to allow learnable weight across heads. Transformers with THSA, named *Tunable Transformers (Tuformers)*, have several advantages compared against vanilla Transformers: **(1)** A guaranteed higher expressive power. We prove that MHSA is a special case of THSA. **(2)** The number of heads is trainable. The concept of the number of heads in THSA generalizes to the stable rank of the core matrix, allowing data-driven implicit training. **(3)** Tuformers allow initialization from pre-trained Transformers such as BERT (Devlin et al., 2019) and its variants.

More importantly, we **formulate a flexible design space** for MHSA where we explore the trade-off between generalization and efficiency, which can be adjusted by the expressive power if there is no over-fitting. Users can choose structures with either higher model efficiency or better generalization that satisfy their own needs. The proposed design space further allows improving the expressive power of the network by finding better tensor representations of the weights through tensor representation theory in future followup works.

We experiment Tuformers with several tasks across multiple domains, from language modeling, machine translation to image generation under a wide range of model sizes. We demonstrate competitive results not only on Tuformers but also in cases where Tuformers are initialized with pre-trained Transformers for other downstream tasks, when combined with Linear Transformer (Katharopoulos et al., 2020) on the image generation task and their efficient variants in the design space.

**Summary of Contributions:**
**(1)** We propose a mathematically rigorous and semantically intuitive *tensor diagram* representation of the multi-head self-attention, introducing a new tool to the ML community for future studies on interpretation and improvements of Transformers.
**(2)** We propose a novel design of the MHSA, *Tunable-Head Self-Attention (THSA)*, resulting in *Tuformers*. Tuformers are structurally more expressive and are showing improved generalization error experimentally.
**(3)** We formulate a flexible design space for attention unit design where users can choose to design structures, depending on the needs, that have better generalization or efficiency.

## 2 TENSOR DIAGRAM REPRESENTATION

**Notations.** We use lower case letters (e.g., $v$) to denote vectors, upper case letters (e.g., $M$) to denote matrices, and curly letters (e.g., $\mathcal{T}$) to denote general tensors. For a tensor $\mathcal{T} \in \mathbb{R}^{I_1 \times \cdots I_M}$, we refer to the number of indices as *order*, each individual index as *mode*, and the length of one mode as *dimension*. For instance, $\mathcal{T}$ is an $M^{\text{th}}$ order tensor that has dimension $I_m$ at its $m^{\text{th}}$ mode. We reserve superscripts to distinguish similar arrays (e.g., $W^Q, W^K, W^V$ are query/key/value weight

matrices), and subscripts to index the elements in an array (e.g., $\boldsymbol{W}_{ij}$ is the $(i, j)^{\text{th}}$ element of $\boldsymbol{W}$). We use colon **:** to slice an array (e.g., $\boldsymbol{W}_{i,:}$ denotes the $i^{\text{th}}$ row of $\boldsymbol{W}$).

We propose to use *tensor diagrams* (Penrose, 1971), a commonly used rigorous/precise and intuitive graphical representation for multi-linear operations among higher-order arrays (i.e., tensors), to represent multi-head self-attention (MHSA). Since MHSA consists of multilinear operations and a nonlinear softmax function, we will introduce tensor diagrams and our novel design extending tensor diagrams to denote the composition of nonlinear and multilinear operations in MHSA.

## 2.1 Tensor Diagram Basics

**Arrays denoted as nodes with legs.** An array is represented as a node with leg(s) in a tensor diagram as shown in Figure 1. We denote the *order* (the number of dimensions) of the array by the number of legs extending from the node. Each labeled leg represents one *mode* of a tensor. Every mode of the tensor needs to be uniquely labeled. We usually use the dimension of the mode as the label (i.e., an associated positive integer written on top of each leg). The legs

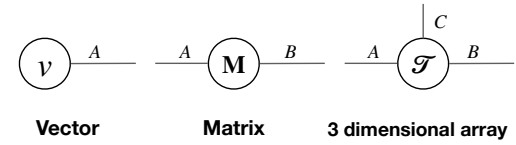

**Figure 1: Arrays in tensor diagram.** A vector is denoted as a node with 1 leg, a matrix as a node with 2 legs and an $N$-dimensional array as a node with $N$ legs.

do not need to be straight lines, and their orientations do not matter. Matrices $\boldsymbol{M} \in \mathbb{R}^{A \times B}$ and $\boldsymbol{M}^{\top} \in \mathbb{R}^{B \times A}$ can be represented via the same tensor diagram as long as the $\boldsymbol{M}$ node has two legs (to denote that it is a matrix, wherever the legs extend to), labeled as $A$ and $B$ (to denote its size).

$$\sum_{k=1}^{K} \boldsymbol{A}_{mk} \boldsymbol{B}_{kn} = \boldsymbol{C}_{mn} \qquad \frac{\exp(\alpha \cdot \boldsymbol{A}_{mn})}{\sum_{n'=1}^{N} \exp(\alpha \cdot \boldsymbol{A}_{mn'})} = \boldsymbol{C}_{mn} \qquad \boldsymbol{A}_{mr} \boldsymbol{B}_{nr} = \boldsymbol{\mathcal{C}}_{nmr}$$

**Figure 2: Tensor diagrams for atomic operations. (1) (left)** Contraction is an operation that generalizes the matrix multiplication. It sums element-wise products on a mode in object $\boldsymbol{A}$ and a corresponding mode (with the same dimension) in object $\boldsymbol{B}$ (i.e., along a leg in node $\boldsymbol{A}$ and a corresponding leg in node $\boldsymbol{B}$). In the tensor diagram, multiplying two matrices (or higher-order tensors with more than 2 legs) corresponds to *"gluing" their corresponding legs (on a certain mode)*. **(2) (middle)** Softmax is an element-wise exponential function normalized along a certain mode. We propose to denote the $\alpha$-*scaled softmax* function $\text{softmax}(\alpha \boldsymbol{A})$ on $\boldsymbol{A}$ as *a dotted box with a labeled filled ball* (to distinguish itself from tensor objects, i.e., nodes which are blank circles) attached to one leg. **(3) (right)** Batch multiplication is an element-wise product along the connected legs.

**Operations in tensor diagrams.** There are three types of operations in the calculation of MHSA:, *contraction*, *softmax*, and *batch multiplication*, as shown in Figure 2 and explained in its caption.

**Evaluation of tensor diagrams: (1) Evaluation order.** We can evaluate a tensor diagram in any pair-wise order except for the nodes in the softmax box. Since the softmax function is nonlinear, we must first evaluate the nodes in the softmax box with arbitrary order before those outside the box. **(2) Reading a tensor diagram.** We can easily identify the output shape by the dangling edges. For instant, in Section 2.1, there are three dangling legs $M$, $N$, $R$. Thus the output is a $3^{\text{rd}}$ order tensor with dimensions $M$, $N$, $R$. Note that the softmax function does not change the shape of the tensor.

**Advantages of tensor diagrams: (1)** Tensor diagram is *orientation invariant*, meaning that we can represent $\boldsymbol{X}$ and $\boldsymbol{X}^{\top}$ using the same tensor diagram. Thus in multi-head, we obtain a universal graphical representation regardless of whether we represent an embedding of each token in the sequence as rows or columns of the input embedding matrix $\boldsymbol{X}$. **(2)** Multi-linear operations are concisely represented and interpreted. **(3)** The representation is both precise in math and intuitive, and the information flow is clear, making analysis of network structure more accessible. In addition, with the labels of the legs, we can read the model complexity explicitly.

We include a comprehensive introduction to tensor diagram representation in Appendix A.

## 2.2 TENSOR DIAGRAM REPRESENTATION OF MULTI-HEAD SELF-ATTENTION

The core of Transformer models is a *multi-head self-attention* (MHSA) module which allows the model to jointly attend to information from different representation sub-spaces at different positions (Vaswani et al., 2017). To distinguish inputs for query, key, value matrices, we use $\boldsymbol{X}^Q \in \mathbb{R}^{N \times F}$, $\boldsymbol{X}^K, \boldsymbol{X}^V \in \mathbb{R}^{M \times F}$ respectively. The MHSA module outputs a matrix $\boldsymbol{M} \in \mathbb{R}^{N \times F}$ as

$$\boldsymbol{Q}_{[h]} = \boldsymbol{X}^Q \boldsymbol{W}_{[h]}^Q; \ \boldsymbol{K}_{[h]} = \boldsymbol{X}^K \boldsymbol{W}_{[h]}^K; \ \boldsymbol{V}_{[h]} = \boldsymbol{X}^V \boldsymbol{W}_{[h]}^V, \tag{1a}$$

$$\mathbf{head}_{[h]} = \mathrm{softmax}\left(\boldsymbol{Q}_{[h]} \boldsymbol{K}_{[h]}^\top / \sqrt{D}\right) \boldsymbol{V}_{[h]}, \tag{1b}$$

$$\boldsymbol{M} = \left[\mathbf{head}_{[1]}, \ \mathbf{head}_{[2]}, \ \cdots, \ \mathbf{head}_{[H]}\right] \boldsymbol{W}^O. \tag{1c}$$

In the above equations, we see how MHSA computes its output in three *steps*:

*(1a) Linear transformation on input embedding to get latent features.* For each $h$, the query/key/value matrices $\boldsymbol{Q}_{[h]}$, $\boldsymbol{K}_{[h]}$, $\boldsymbol{V}_{[h]}$ are linear transformations, parameterized by $\boldsymbol{W}_{[h]}^Q$, $\boldsymbol{W}_{[h]}^K$, and $\boldsymbol{W}_{[h]}^V \in \mathbb{R}^{F \times D}$ respectively, of the input embedding.

*(1b) Scaled dot-product attention.* Each $\mathbf{head}_{[h]} \in \mathbb{R}^{N \times D}$ computes a scaled dot-product between a latent feature matrix $\boldsymbol{Q}_{[h]} \in \mathbb{R}^{N \times D}$ (i.e., a *query* matrix) and another latent feature matrix $\boldsymbol{K}_{[h]} \in \mathbb{R}^{M \times D}$ (i.e., a *key* matrix) along $D$, and applies a softmax function to obtain the weights to be multiplied with a third latent feature matrix $\boldsymbol{V}_{[h]} \in \mathbb{R}^{M \times D}$ (i.e., a *value* matrix).

*(1c) Concatenation and contraction for MHSA.* After concatenation of the $H$ heads, a linear transformation of the concatenated heads, parameterized by $\boldsymbol{W}^O \in \mathbb{R}^{HD \times F}$, is implemented to get the outcome $\boldsymbol{M}$ of the MHSA as a $\mathbb{R}^{N \times F}$ object.

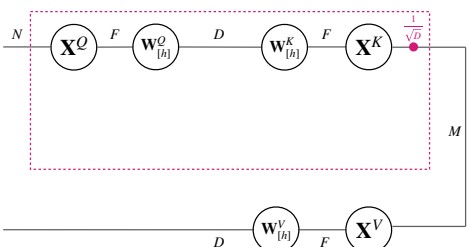
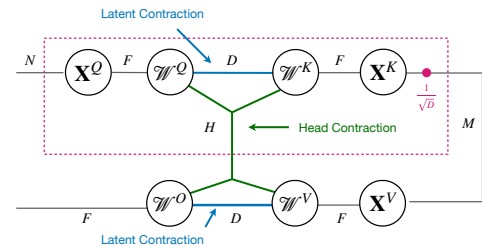

**(a)** Single-head Self-Attention  **(b)** Multi-Head Self-Attention

**Multi-head self-attention in tensor diagram.** The *concatenation* of $H$ matrices $\{\mathbf{head}_{[h]}\}_{h=1}^H$ can be viewed as constructing a three dimensional array with the third mode being the number of heads, i.e., three-legged nodes in tensor diagram notation. This way, the weight matrices $\{\boldsymbol{W}_{[h]}^Q\}_{h=1}^H$ are stacked to be an order 3 tensor $\boldsymbol{\mathcal{W}}^Q$. It is the same with other weight matrices. Since in MHSA, the attention calculation among key, query and value matrices are done separately within each head, batch multiplication along mode $H$ is a proper way to denote these "head-wise" operations. Finally, we connect the resultant tensor with $\boldsymbol{\mathcal{W}}^O$ to obtain MHSA as in Figure 3b. See Appendix B for more details and the proof.

# 3 TUFORMERS: DATA-DRIVEN DESIGN WITH HIGHER EXPRESSIVE POWER

## 3.1 MOTIVATION: INTERPRETATION OF MHSA

In MHSA, $\boldsymbol{Q}_{[h]}$ and $\boldsymbol{K}_{[h]}$ can only contribute to $\mathbf{head}_{[h]}, \forall h$. In order to characterize the information flow in MHSA, we stack the weight matrices $\boldsymbol{W}_{[h]}^Q, \forall h$ together (similarly for $\boldsymbol{W}_{[h]}^K$ and $\boldsymbol{W}_{[h]}^V$) and introduce a matrix $\boldsymbol{C}$ to interpret MHSA. Guided by its tensor diagram representation in Figure 3b, $\boldsymbol{C}$ is clearly characterized as in Proposition 1.

**Proposition 1** (Interpretation of MHSA). *With weight matrices stacked as* $\overline{\boldsymbol{W}}^Q := [\boldsymbol{W}_{[1]}^Q, \cdots, \boldsymbol{W}_{[H]}^Q], \overline{\boldsymbol{W}}^K := [\boldsymbol{W}_{[1]}^K, \cdots, \boldsymbol{W}_{[H]}^K], \overline{\boldsymbol{W}}^V := [\boldsymbol{W}_{[1]}^V, \cdots, \boldsymbol{W}_{[H]}^V] \in \mathbb{R}^{F \times DH},$ *and a core matrix* $\boldsymbol{C} := \boldsymbol{I}_H \otimes (\mathbf{1}_D \mathbf{1}_D^\top) \in \mathbb{R}^{DH \times DH}$ *( i.e., a Kronecker product of an all-ones matrix* $\mathbf{1}_D \mathbf{1}_D^\top \in \mathbb{R}^{D \times D}$ *and an identity matrix* $\boldsymbol{I}_H \in \mathbb{R}^{H \times H}$ *), MHSA is equivalent to*

$$\overline{Q} = X^Q \overline{W}^Q; \; \overline{K} = X^K \overline{W}^K; \; \overline{V} = X^V \overline{W}^V, \tag{2a}$$

$$\overline{\textbf{head}}_r = \text{softmax}\left(\sum_{s=1}^{DH} C_{rs} \, \overline{Q}_s \overline{K}_s^\top / \sqrt{D}\right) \overline{V}_r, \tag{2b}$$

$$M = \left[\overline{\textbf{head}}_1, \overline{\textbf{head}}_2, \; \cdots, \; \overline{\textbf{head}}_{DH}\right] W^O. \tag{2c}$$

*In tensor diagram notation, MHSA is equivalent to Figure 4a. Here, we use $\overline{Q}_r, \overline{K}_r, \overline{V}_r \in \mathbb{R}^F$, $\overline{\textbf{head}}_r \in \mathbb{R}^N$ to denote the $r^{\text{th}}$ column of the corresponding matrices, and $C_{rs}$ to denote the $(r, s)^{\text{th}}$ element of $C$.*

Proposition 1 is proved in Appendix D.1. Note that $\{\overline{\textbf{head}}_1, \cdots, \overline{\textbf{head}}_D\}$ correspond to $\textbf{head}_{[1]}$, $\{\overline{\textbf{head}}_{D+1}, \cdots, \overline{\textbf{head}}_{2D}\}$ correspond to $\textbf{head}_{[2]}$ and so forth. The intuition of interpreting MHSA using Proposition 1 comes from the tensor diagram graphical representation of MHSA structure. From the tensor diagram of MHSA as shown in Figure 3b, we see two types of contractions: latent contractions along $D$ and head contractions along $H$. The former corresponds to $(\mathbf{1}_D \mathbf{1}_D^\top)$ part of $C$ and the latter corresponds to $I_H$ part of $C$ as shown in Figure 4a.

**Motivation for an improved design: from MHSA (a preset $C$) to THSA (a learnable $C$).** In MHSA, with the preset specific $C = I_H \otimes (\mathbf{1}_D \mathbf{1}_D^\top)$, it restricts the possible weight sharing. In other words, $Q_{[h]}$ and $K_{[h]}$ can only contribute to $\textbf{head}_{[h]}, \forall h$. However, there is no motivation why such a preset and fixed $C$ is desirable or why we do not allow $Q_{[h]}$ or $K_{[h]}$ to contribute to other heads. Our motivation for an improved design is simple: rather than presetting the core matrix as $C = I_H \otimes (\mathbf{1}_D \mathbf{1}_D^\top)$, we allow a data-driven learnable $C$, and we call such a design *Tunable-head self-attention (THSA)*. Now with the learnable $C$ in THSA, we allow $Q_{[h]}$ and $K_{[h]}$ to contribute to other heads $\textbf{head}_{[h']}$, where $h' \neq h$. In other words, THSA allows learnable weights across heads.

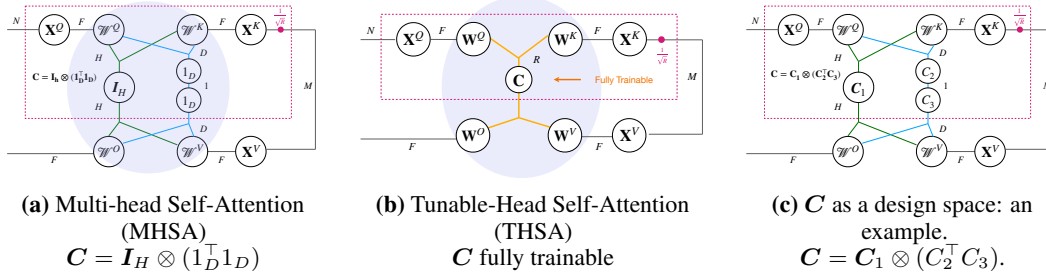

**(a)** Multi-head Self-Attention (MHSA)
$C = I_H \otimes (\mathbf{1}_D^\top \mathbf{1}_D)$

**(b)** Tunable-Head Self-Attention (THSA)
$C$ fully trainable

**(c)** $C$ as a design space: an example.
$C = C_1 \otimes (C_2^\top C_3)$.

**Figure 4:** MHSA as a special case of THSA. Figure 4a (left) represents MHSA. Figure 4b (middle) represents THSA. As shown in Proposition 1, weight matrices in MHSA collaborate in a very specific way. When $C = I_H \otimes (\mathbf{1}_D^\top \mathbf{1}_D)$ in Proposition 1, THSA degrades to to MHSA. Therefore, it is natural to make $C$ fully trainable for higher expressive power. THSA also reveals a design space. For example, we can extend MHSA by allowing trainable components $C_1$, $C_2$ and $C_3$ as in Figure 4c.

## 3.2 A NOVEL DESIGN: TUNABLE-HEAD SELF-ATTENTION (THSA)

As shown in Figure 4b, a Tunable-head self-attention (THSA) [2] module has the same input and output domains as an MHSA, i.e., it takes the input embedding $X^Q \in \mathbb{R}^{N \times F}$ and $X^K, X^V \in \mathbb{R}^{M \times F}$ as inputs and returns a matrix $T \in \mathbb{R}^{N \times F}$ as output. A THSA, with five weight matrices $W^Q, W^K, W^V \in \mathbb{R}^{F \times R}, W^O \in \mathbb{R}^{R \times F}$, and $C \in \mathbb{R}^{R \times R}$, is mathematically described as

$$Q = X^Q W^Q; \; K = X^K W^K; \; V = X^V W^V, \tag{3a}$$

$$\Omega_r = \text{softmax}\left(\sum_{s=1}^{R} C_{rs} \, Q_s K_s^\top / \sqrt{R}\right) V_r, \tag{3b}$$

$$T = \Omega W^O = [\Omega_1, \Omega_2, \; \cdots, \; \Omega_R] W^O. \tag{3c}$$

The intermediate result $\Omega_r \in \mathbb{R}^N$ is the $r^{\text{th}}$ column of the matrix $\Omega$. In MHSA, the softmax function is scaled, which normalizes its input by $\sqrt{D}$, i.e., the square root of the latent dimension. The scaling

---

[2]THSA improves MHSA and only modifies the intra-layer structure. It does not change the inter-layer structure outside MHSA.

ensures that the distribution of the attention matrix approximates a standard Gaussian distribution. To achieve a similar effect, in THSA, we change the constant to $\sqrt{R}$ accordingly.

**THSA generalizes MHSA.** THSA has learnable $C$. If $C = I_H \otimes (1_D 1_D^\top)$, i.e., a Kronecker product of an all-ones matrix $1_D 1_D^\top \in \mathbb{R}^{D \times D}$ and an identity matrix $I_H \in \mathbb{R}^{H \times H}$, the THSA reduces to an MHSA with $H$ heads and latent dimension $D$, as illustrated in Figure 4a. We call a Transformer with THSA module as Tunable Transformers (Tuformers). The tensor diagram representation of THSA is given in Figure 4b. It has an learnable core matrix $C$ compared to MHSA in Figure 4a which has a preset core matrix.

**Tuformers can be initialized with Pre-trained Transformers.** Although different in the structure, Tuformers can always be initialized with pre-trained Transformer models through re-parameterization. MHSA is updated by weight matrices $\{W_{[h]}^K\}_{h=1}^H$, $\{W_{[h]}^Q\}_{h=1}^H$, $\{W_{[h]}^V\}_{h=1}^H$ and $W^O$ while in THSA we have $W^Q$, $W^K$, $W^V$, $W^O$ and $C$. To initialize THSA with MHSA, we follow the scheme proposed in Proposition 1. $C$ is the kronecker product between a $D \times D$ all-one matrix and an $H \times H$ identity matrix, where $D$ is the latent dimension and $H$ is the number of heads in MHSA. $W^O$ remains the same as that of MHSA. As for $W^Q$, $W^K$ and $W^V \in \mathbb{R}^{F \times R}$, we concatenate $\{W_{[h]}^K\}_{h=1}^H$, $\{W_{[h]}^Q\}_{h=1}^H$, $\{W_{[h]}^V\}_{h=1}^H \in \mathbb{R}^{F \times D}$ along mode $D$.

**The notion of heads in THSA.** Due to the learnable $C$ and data-dependent weight sharing mechanism, the notion of heads in THSA is not as obvious as in MHSA. Although we have an upper bound $R$ of the number of heads, which is achieved when the learned $C$ decides not to share any weights across heads, the actual effective number of heads depends on $C$. As a result, we define the number of heads in THSA as the stable rank of the core matrix $C$, which is $\sum_i \sigma_i^2 / \max_i \sigma_i$, where $\sigma_i$ is the singular value of $C$. We use the stable rank because it is largely unaffected by tiny singular values.

**THSA as a flexible design space:** $C$ in THSA ( Figure 4b ) naturally reveals a design space. In MHSA, $C$ is completely fixed and in THSA $C$ is fully trainable. We can also have flexible $C$ designs for user-specific needs of either increasing generalization or improving efficiency.

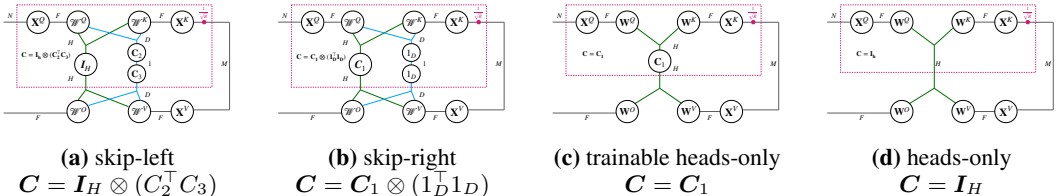

| **(a)** skip-left | **(b)** skip-right | **(c)** trainable heads-only | **(d)** heads-only |
|:---:|:---:|:---:|:---:|
| $C = I_H \otimes (C_2^\top C_3)$ | $C = C_1 \otimes (1_D^\top 1_D)$ | $C = C_1$ | $C = I_H$ |

**Figure 5:** $C$ naturally reveals a design space. We can obtain better generalization or better efficiency by allowing fully trainable $C$, partially trainable $C$, pre-set fixed $C$ and etc. We demonstrate some of them extending from the structure of MHSA where $C$ takes the form of a Kronecker product as shown in Figure 4c. The relationship between the generalization and efficiency of these designs is summarized in Figure 6.

**Better generalization.** Better generalization can be obtained by increasing the expressive power if there is no over-fitting. A naive way to increase the expressive power is to add more parameters in the weight matrices in MHSA, such as increasing the latent dimension $D$ or the number of heads $H$. We find that a more effective way to increase the expressive power for better generalization is to find more expressive structures. We demonstrate a few of them in Figure 5 and Figure 4b. In MHSA, $C$ is pre-set fixed where $C = I_H \otimes (1_D^\top 1_D)$. In Figure 5a (skip-left $C = I_H \otimes (C_2^\top C_3)$ ) and Figure 5b (skip-right $C = I_H \otimes (C_2^\top C_3)$ ), we maintain the form of a Kronecker product but allow either $C_1$ or $C_2$, $C_3$ trainable. We can think of skip-left and skip-right as skip-connection designs in $C$. In THSA, $C$ is fully trainable. It is easy to observe an increasing in expressive power from MHSA to skip-left, skip-right, and finally THSA, together with an increasing in generalization in the same order. Our empirical study

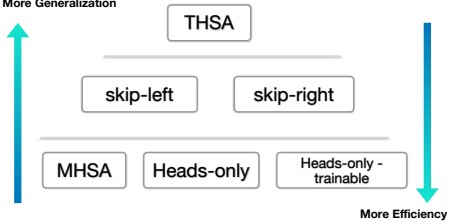

**Figure 6:** We can divide all structures listed in Figure 4 and Figure 5 into different groups by their expressive power. The higher group a structure is in, the higher expressive it has. Better generalization can be obtained in more expressive structures. We can also go for very efficient ones if we are willing to sacrifice some generalization. More details in Section 4.

also show that these structurally more expressive models more effectively improve the generalization than naively increase the number of parameters.

**More Efficiency.** We can obtain different levels of efficient structures in the design space by adjusting the expressive power. Compared to MHSA, skip-left and skip-right have more generalization in various tasks but almost the same number of parameters. Being special cases of skip-right, Figure 5c ( trainable heads-only $C = C_1$ ) and Figure 5d ( heads-only $C = I_H$ ) are less expressive. However, trainable heads-only only has around $6\%$ of the parameters compared to MHSA while heads-only only has $1.5\%$ with most performance maintained.

There is a trade-off between generalization and efficiency in the design space $C$ as demonstrated in Figure 6. We empirically verify that without over-fitting, generalization and efficiency have similar hierarchy as expressive power. Thus we can design structures in different level of expressive power, tailoring needs for either better generalization or more efficiency. THSA has the best generalization, but $125\%$ parameters compared to MHSA. Since skip-left and skip-right are special cases of THSA, it is natural that their expressive power and generalization are lower. However, they are more efficient. With almost the same number of parameters, they have better generalization than MHSA. Heads-only and trainable heads-only are very efficiency sacrificing an acceptable amount generalization. More details can be found in Section 4.

## 4 EXPERIMENTS

**Datasets, tasks and evaluation metrics.** We evaluate Tuformers on 7 datasets for 5 tasks: word-level Language Modeling (LM) on Penn Treebank (PTB) (Marcus et al., 1993), Neural Machine Translation (NMT) on WMT16 'English-German' (Sennrich et al., 2016), Automatic Speech Recognition (ASR) on LibriSpeech (Panayotov et al., 2015), Natural Language Inference (NLI) on MNLI (Williams et al., 2018) and QNLI (Wang et al., 2018) and Image Generation on CIFAR10 (Krizhevsky, 2009) and MNIST (Deng, 2012) datasets.

**Baselines.** We compare the performance of MHSA module with THSA module in the following 5 backbone architectures: **(1)** a vanilla Transformer model (Vaswani et al., 2017), **(2)** an Transformer-XL model (Dai et al., 2019), **(3)** a BERT-large model (Devlin et al., 2019), **(4)** a RoBERTa-large model (Liu et al., 2019), and **(5)** an ALBERT model (Lan et al., 2019). In all baselines, we adopt all default settings used in the original paper (hyperparameters specified in Appendix E), and only replace/re-parameterize the MHSA to THSA.

### 4.1 EMPIRICAL RESULTS OF TUFORMERS.

In this part, we first show that THSA (used in Tuformers) universally outperforms MHSA (used in Transformers) diverse tasks under different model scales and THSA is simple to use in practice.

*(1) THSA consistently outperforms MHSA on diverse datasets for a variety of tasks under different model scales.* In LM, NMT and ASR tasks, we compare THSA against the MHSA models under different model scales. MHSA is parameterized by four weight matrices $W^Q$, $W^K$, $W^V$ and $W^O$ whereas THSA adds an additional core matrix $C$ of the same size. Adopting the size of the four original weight matrices in MHSA to THSA incurs an $25\%$ increase in the number of parameters. For a fair comparison, we first compare MHSA against THSA with the same number of parameters (thus with smaller query/key/value/output weight matrices). To get a full picture, we also compare MHSA against THSA with the same sized four original weight matrices. As shown in Figure 7, under the same number of parameters, our THSA (blue curve) universally outperforms MHSA (red curve) under all model scales. By increasing the number of parameters of the attention units by 25% [3] (i.e., adding the core matrix $C$ while keeping the size of the query/key/value/output weight matrices the same), our THSA (purple curve) further improves the performance universally under all model scales.

*(2) Tuformers initialized with pre-trained strong models further improve the performance.*

In NLI task, we initialize Tuformers with pre-trained BERT-large, RoBERTa-large and ALBERT models, and perform the same fine-tuning tasks. As shown in Table 1, THSA is compatible with

---

[3]As will be discussed later, the overall model size is not drastically increased.

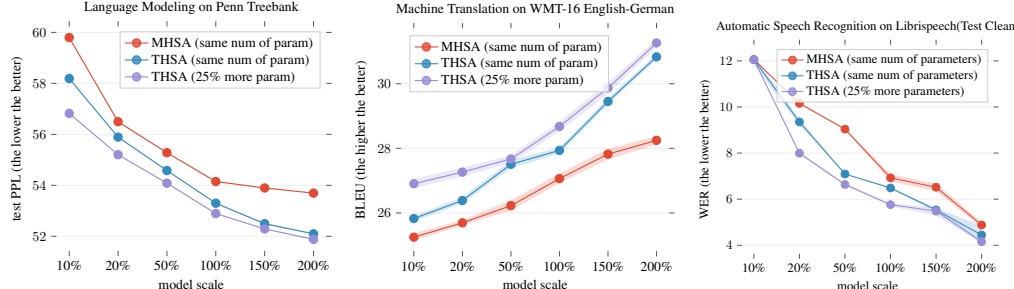

**Figure 7:** Performance comparison of THSA and MHSA in **(left)** language modeling on Penn Treebank dataset on TransformerXL, **(middle)** neural machine translation on WMT16 English-German dataset and **(right)** automatic speech recognition on Librispeech on Transformers. Under the same number of parameters, our THSA universally outperforms MHSA under all model sizes. The error bars displayed in shades are generated using 10 random runs in LM and NMT and 3 random runs in ASR.

**Table 1: (Left) THSA initialized with pre-trained models.** Comparison of accuracies obtained for language inference task on MNLI and QNLI datasets, the higher the better. Results show that Tuformers can be conveniently initialized with pre-trained models for downstream tasks, and fine-tuning obtains impressive performance. **(Right) THSA +kernels.** Comparison of bits per dimension (BPD) for image generation task on MNIST and CIFAR-10 datasets, the lower the better. Results show that Tuformers can be extended to image generation tasks and improve the performance of other efficient designs with linear computational and memory complexities.

| Models | MNLI | QNLI |
|---|---|---|
| BERT (Devlin et al., 2019) | 84.1% | 92.1% |
| BERT + THSA | **84.9%** | **92.8%** |
| RoBERTa (Liu et al., 2019) | 89.4% | 94.3% |
| RoBERTa + THSA | **90.2%** | **94.7%** |
| ALBERT (Lan et al., 2019) | 89.1% | 97.9% |
| ALBERT + THSA | **89.5%** | **98.1%** |

| Models | MNIST | CIFAR 10 |
|---|---|---|
| ELU kernel (Katharopoulos et al., 2020) | 0.72 | 3.51 |
| ELU kernel + THSA | **0.65** | **3.42** |
| Polynomial kernel (Tsai et al., 2019) | 0.64 | 3.47 |
| Polynomial kernel THSA | **0.57** | **3.37** |

existing pre-trained Transformer models and there is no need to train Tuformers from scratch: fine-tuning THSA initialized from pre-trained BERT and its variants further improves their performance.

*(3) Combing THSA with kernels to obtain linear computation and memory complexities in sequence length.* Tuformers can also be incorporated into efficient variants of Transformers for lower computation and memory complexities. We experiment on image generation tasks on CIFAR10 (Krizhevsky, 2009) and MNIST (Deng, 2012) datasets as in Table 1 and show that Tuformers obtain strong performance and linear space complexity when combined with kernels.

## 4.2 FLEXIBLE DESIGN SPACE

*(1) THSA reveals a flexible design space $C$ for Transformer.* We empirically demonstrate that there exists a trade-off between generalization and efficiency in this design space, which could be adjusted by the expressive power. By making $C$ fully-trainable, partially trainable or pre-set fixed as shown in Figure 5 and Figure 4, we demonstrate that some structures in $C$ improves the generalization while some other designs improve the efficiency of MHSA.

*(2) Models with higher expressive power in the design space have better generalization.* As shown in Table 2 and Figure 7, we compare two ways that increase the expressive power for better generalization: naively increasing the number of parameters and designing more expressive structures. Both approaches increase generalization. However, naively increasing the number of parameters is not as effective as adopting more expressive structures. With the same number of parameters, more expressive structures, such as THSA, skip-left and skip-right, universally outperform MHSA.

**Table 2:** Performance in the flexible design space.

| Models | NMT (BLEU) | # params | Train | Test |
|---|---|---|---|---|
| MHSA | 28.9 | 100% | 10.6 | 3.2 |
| MHSA with 125% params | 29.0 | 125% | 11.2 | 3.6 |
| MHSA with 1.5% params | 20.2 | 1.5% | 6.4 | 2.5 |
| THSA | 29.4 | 125% | 11.5 | 3.9 |
| skip-right $C = C_1 \otimes (1_D^\top 1_D)$ | 29.3 | 101% | 10.6 | 3.2 |
| skip-left $C = I_H \otimes (C_2^\top C_3)$ | 29.2 | 100% | 10.6 | 3.2 |
| trainable heads-only $C = C_1$ | 27.4 | 6% | 7.4 | 2.8 |
| heads-only $C = I_H$ | 27.1 | 1.5% | 6.1 | 2.3 |

*(3) Models can also be made very efficient.* Moreover, in Figure 5c ( trainable heads-only $C = C_1$ ), we trade generalization for efficiency. With $6\%$ parameters, we maintain $95\%$ performance in the NMT task. In a more extreme case, if we only maintain the head contraction as shown in Figure 5d, the number of parameters goes down to $1.5\%$ with a $42\%$ faster training and $30\%$ faster testing. Surprisingly, almost $94\%$ performance is obtained with this light-weighted model.

### 4.3 ABLATION STUDY: CHECK-POINTING TO ALLEVIATE MEMORY OVERHEAD WITHOUT KERNELS.

Although an effective design with guaranteed higher expressive power, Tuformers come with an extra memory overhead when calculating values within the softmax. In Tuformers, if done naively, the memory overhead is $MNR$ where $N, M$ denote the sequence length and $R = DH$ is the rank of the core tensor, incurring a $D$ times larger memory overhead than vanilla Transformers in calculating softmax. Combining Tuformers with some state-of-the-art kernel-based efficient Transformers such as Linear Transformers using ELU (Katharopoulos et al., 2020) and polynomial (Tsai et al., 2019) kernels, we reduce the computation and memory complexities of Tuformers to be linear in the sequence length by removing the nonlinearity constrain in the softmax.

We can also alleviate the $D$ times larger memory overhead compared with vanilla Transformers in calculating softmax without using kernels. Specifically, we use check-pointing to eliminate the memory overhead in THSA. Since no intermediate results are saved in the forward pass using check-pointing, we recalculate the intermediate results of the softmax box in the back-propagation, which introduces some computational overhead. We observe that applying check-pointing to Tuformers leads to only a slight increase in training time while relieving the memory overhead. We can reduce or even avoid the repetitive calculations by developing a proper backward module, although such development is beyond the scope of this paper and deferred for future works. See Figure 12 for more details.

## 5 RELATED WORKS

**Analysis of MHSA.** Given the popularity of Transformers in a wide range of domains, a line of works focuses on understanding and improving MHSA. Voita et al. (2019) proposed a practical scheme to prune less informative heads. Michel et al. (2019) also showed that most attention heads are removable in testing without much performance compromise. Cordonnier et al. (2020) found empirical evidence of redundancy in key/query projections and proposed a re-parameterization scheme. These works bring valuable insights into MHSA, but a consistent theoretical analysis for these findings is missing. Our work differs from these works in that we focus on the theoretical understanding of MHSA in terms of expressive power, and we provide theoretical guarantees for a novel design, Tuformers, with higher expressive power and data-driven trainable heads.

**Efficient Transformers.** MHSA has a global receptive field, i.e., each token's representation is updated by attending to all other tokens, therefore incurring a quadratic memory and computation complexities concerning the sequence length. An extensive line of works focusing on reliving such the dependencies on the sequence length, such as Performer (Choromanski et al., 2020), Reformer (Kitaev et al., 2020), Linformer (Wang et al., 2020) and Linear Transformer (Katharopoulos et al., 2020). Our proposed model, Tuformers, can be incorporated into some kernel-based efficient Transformers, which applies kernel-based methods to MHSA to remove the nonlinearity constraint brought by the softmax function, achieving the same state-of-the-art linear computation and memory complexities in the sequence length.

## 6 CONCLUSION

This paper introduces a mathematically rigorous yet intuitive tensor diagram representation of MHSA, formulates a design space where we can analyze the expressive power of MHSA and its variants and proposes Tuformers, a novel model design with a guaranteed higher expressive power. Furthermore, Tuformers have a data-driven structure where heads can be trained implicitly and initialized with pre-trained Transformer models. The introduction of the tensor diagram representations and the theory presented in this paper provide new tools and open new directions for future research searching for expressive and efficient architectures.

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

## A  Supplementary Material for Tensor Diagrams

In this section, we provide a comprehensive introduction to the tensor diagram representation.

### A.1  Components of Tensor Diagram: Arrays as Nodes with Leg(s)

**Arrays denoted as Nodes with Legs.** An array is represented as a node with leg(s) in a tensor diagram as shown in Figure 1. The *order* (the number of dimensions) of the array is denoted by the number of legs extending from the node. The legs do not need to be straight lines ,and the orientation does not matter. For example, matrices $A$ and $A^\top$ are equivalent in this notation. Each labeled leg represents one *mode* of a tensor. Every mode of the tensor needs to be uniquely labeled. We usually use the dimension of the mode as the label (i.e., an associated positive integer written on top of each leg). If multiple modes of the same tensor have the same length, we use subscripts to differentiate them.

### A.2  Operations in Tensor Diagram

$$\sum_{k=1}^{K} A_{mk} B_{kn} = C_{mn} \qquad \frac{\exp(\alpha \cdot A_{mn})}{\sum_{n'=1}^{N} \exp(\alpha \cdot A_{mn'})} = C_{mn} \qquad A_{nr} B_{mr} = \mathcal{C}_{nmr}$$

**Figure 8: (1) Contraction** as shown in Appendix A.2 is an operation that generalizes the matrix multiplication. It does summation on element-wise products along a mode in object 1 and a corresponding mode (with the same dimension) in object 2 (i.e., in tensor diagram language, along a leg in node 1 and a corresponding leg in node 2). **In tensor diagram, multiplying two matrices (or higher-order tensors with more than 2 legs) corresponds to "gluing" their corresponding legs (along a certain mode). (2) Softmax** as shown in Appendix A.2 is an element-wise exponential function normalized along a certain mode. Tensor diagram has a convention to denote the contraction, but not the softmax. We propose to denote the $\alpha$-**scaled softmax** function $\mathrm{softmax}(\alpha A)$ on $A$ as **a dotted box with a labeled filled ball** (to distinguish itself from tensor objects, i.e., nodes which are blank circles) attached to one leg. **(3) Batch Multiplication**, as shown in appendix A.2, is an elementwise product along the connected legs.

There are three types of operations:*contraction*, *softmax* and *batch multiplication*. *Contraction* is an operation that does summation on element-wise products along a mode in object 1 and a corresponding mode (with the same dimension) in object 2 (i.e., in tensor diagram language, along a leg in node 1 and a corresponding leg in node 2), whereas *softmax* is an element-wise exponential function normalized along a certain mode. Tensor diagram has a convention to denote the contraction, but not the softmax. We will first introduce contraction in tensor diagram notation and then propose our design of softmax in tensor diagram language.

**Contractions denoted as edges connecting the node legs.** In tensor diagram, multiplying two matrices (or higher-order tensors with more than 2 legs) corresponds to "gluing" their corresponding legs (along a certain mode) as shown in Appendix A.2, and it is called *tensor contraction*. Since the representation of arrays in the tensor diagram is orientation invariant, contractions are also orientation invariant. This is especially useful in representing the self-attention unit. In some cases, the data object are represented as row vectors while in some other scenarios they are denoted as column vectors. If we use mathematical formula, we need to take care of the ordering of the matrix multiplication because $A^\top B, AB^\top, BA^\top, B^\top A$ and more are all different. In contrast, tensor diagram provides a universal graphical representation as long as the corresponding legs are connected correctly.

**Nonlinear activation denoted as dotted box with a labelled filled ball attached.** We propose to denote the $\alpha$-scaled softmax function $\mathrm{softmax}(\alpha A)$ on $A$ as a dotted box with a labeled filled ball (to distinguish itself from tensor objects, i.e., nodes which are blank circles) attached to one leg. The label right above the filled ball denote the scaling parameter $\alpha$, and the leg that the ball is attached to

indicate the mode where normalization is implemented along in the softmax operation. As shown in Appendix A.2, let $A$ denote the resultant of some operations (for example $Q_{[h]}K_{[h]}^\top$), the $\frac{1}{\sqrt{D}}$-scaled softmax on $A$ which normalize along leg $N$ is depicted. Softmax does not change the shape of the input. That is why we introduce a dotted box, which does not change the shape either, to denote it.

**Batch Multiplication** Given $A$ and $B$, the batch multiplication is an element-wise product along the connected legs. The resultant tensor $C$ is obtained by merging connected nodes, maintaining the dangling legs $M$, $N$ and $R$. Since in MHSA, the attention calculation among key, query and value matrices are done separately within each head, batch multiplication is a proper way to denote these "head-wise" operations.

## A.3 ADVANTAGES OF TENSOR DIAGRAM

Tensor diagram representations enjoy a few advantages compared with using flow charts together with math formulas.

**(1) The notation is orientation invariant.** The input embedding matrix $X$ can be represented as either $\mathbb{R}^{N \times F}$ or $\mathbb{R}^{F \times N}$ with rows or columns being the embedding of each token in the sequence respectively. Although we take the former convention in our mathematical notations in Equation B.1, there are papers that use the latter, creating discrepancies between notations. However, in tensor diagram, since the node legs are orientation invariant, we obtain a universal graphical representation of multi-head self-attention, irrespective of how the input embedding matrix is represented, as shown in Figure 9c.

**(2) The representation is both precise in math and intuitive, making analysis easier.** The tensor diagram itself is mathematically rigorous.

With the labels of the legs, the model complexity (number of parameters) is explicitly displayed, requiring no supplementary information such as the size of the parameters $W^Q$, $W^K$ and $W^V$ as in mathematical formulas. In addition,the resultant from an operation or a sequence of operations (i.e., a connected sub-graph of the entire tensor diagram graph) can be treated as "merging" any connected nodes in the sub-graph. This is particularly convenient to obtain the size of the resultant: any the dangling leg becomes a leg of the resultant.

**(3) Multi-linear operations are concisely represented and interpreted.** Using tensor diagram, we can concisely represent the multi-head self-attention rigorously in one diagram as shown in Figure 9c. More important, the multi-linear interaction between the latent features $Q_{[h]}$, $K_{[h]}$ and $V_{[h]}$, which essentially attribute to the multi-linear interaction between the weight tensors $W^Q$, $W^K$ and $W^B$, is clearly illustrated in the tensor diagram. This illustration, which is not achieved by any existing representations, is crucial for our principled understanding of the multi-head self-attention.

**(4) The information flow is clear.** For example, in Figure 9a there is no direct information pass from $W_{[h]}^K$, $W_{[h]}^Q$ to $W_{[h]}^V$ while in Figure 9c, such exchange is done through edge $D$s and $H$s. It is such observation (there are multiple paths that connect the weight parameter matrices) that leads to our idea of comparing expressive power between different structures.

# B  SUPPLEMENTARY MATERIAL FOR MULTI-HEAD SELF-ATTENTION

In the section, we prove the equivalence between the tensor diagram and the equations for *multi-head self-attention* (MHSA). To establish such an equivalence, we will need to show that both tensor diagram and the matrix equations lead to the same result in element-wise notation.

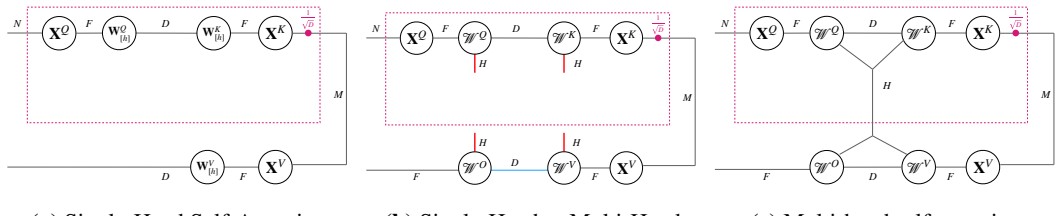

**(a)** Single-Head Self-Attention  **(b)** Single-Head to Multi-Head  **(c)** Multi-head self-attention

**Figure 9: Tensor diagrams from single-head self-attention to multi-head self-attention.** Figure 9a is the tensor diagram representation of a single-head self-attention. Figure 9c is the tensor-diagram representation of a multi-head self-attention.

For convenience of reference, we recap both representations in Equation (B.1) and Figure 9.

$$\boldsymbol{Q}_{[h]} = \boldsymbol{X}^Q \boldsymbol{W}_{[h]}^Q; \ \boldsymbol{K}_{[h]} = \boldsymbol{X}^K \boldsymbol{W}_{[h]}^K; \ \boldsymbol{V}_{[h]} = \boldsymbol{X}^V \boldsymbol{W}_{[h]}^V. \tag{B.1a}$$

$$\mathsf{head}_{[h]} = \mathrm{softmax}\left(\boldsymbol{Q}_{[h]} \boldsymbol{K}_{[h]}^\top / \sqrt{D}\right) \boldsymbol{V}_{[h]}, \tag{B.1b}$$

$$\boldsymbol{M} = \left[\mathsf{head}_{[1]}, \mathsf{head}_{[2]}, \cdots, \mathsf{head}_{[H]}\right] \boldsymbol{W}^O, \tag{B.1c}$$

Note that the query weight tensor $\mathcal{W}^Q$ comes from a concatenation of $H$ query weight matrices $\{\boldsymbol{W}_{[h]}^Q\}_{h=1}^H$ such that $\mathcal{W}_{h,:,:}^Q = \boldsymbol{W}_{[h]}^Q$. Similarly, $\mathcal{W}^K, \mathcal{W}^V$ are concatenated from $\{\boldsymbol{W}_{[h]}^K\}_{h=1}^H$, $\{\boldsymbol{W}_V[h]\}_{h=1}^H$ such that $\mathcal{W}_{h,:,:}^K = \boldsymbol{W}_{[h]}^K, \mathcal{W}_{h,:,:}^V = \boldsymbol{W}_{[h]}^V$. Furthermore, we denote $\mathcal{Q}, \mathcal{K}, \mathcal{V}$ as the reshaped tensors from $\{\boldsymbol{Q}_{[h]}\}_{h=1}^H$, $\{\boldsymbol{K}_{[h]}\}_{h=1}^H$, $\{\boldsymbol{V}_{[h]}\}_{h=1}^H$. Finally, $\mathcal{W}^O$ is reshaped from $\boldsymbol{W}^O$.

To differentiate between equations and tensor diagrams in the proof, we add an over-line to every object in the tensor diagram throughout this section.

## B.1  SINGLE-HEAD SELF-ATTENTION (SHSA)

We will prove that the tensor diagram in Figure 9a is equivalent to Equations (B.1a) and (B.1b).

*Proof.* For simplicity, we omit the subscript $[h]$ in this subsection.

We first show that a contraction of the nodes within the softmax box is equivalent to $\boldsymbol{Q}\boldsymbol{K}^\top$. Let $\boldsymbol{A} = \boldsymbol{Q}\boldsymbol{K}^\top = \boldsymbol{X}^Q \boldsymbol{W}^Q (\boldsymbol{W}^K \boldsymbol{X}^K)^\top$, where $\boldsymbol{X}^Q \in \mathbb{R}^{N \times F}$, $\boldsymbol{X}^K \in \mathbb{R}^{M \times F}$, $\boldsymbol{W}^Q, \boldsymbol{W}^K \in \mathbb{R}^{F \times D}$. We use subscripts in the element-wise notation to differentiate edges when multiple edges with same dimension appear in the same equation.

$$\boldsymbol{Q}_{nd} = \sum_{f_1=1}^{F} \boldsymbol{X}_{nf_1}^Q \boldsymbol{W}_{f_1 d}^Q, \tag{B.2a}$$

$$\boldsymbol{K}_{md} = \sum_{f_2=1}^{F} \boldsymbol{X}_{mf_2}^K \boldsymbol{W}_{f_2 d}^K, \tag{B.2b}$$

$$\boldsymbol{A}_{nm} = \sum_{d=1}^{D} \boldsymbol{Q}_{nd} \boldsymbol{K}_{dm} = \sum_{d=1}^{D} \sum_{f_1=1}^{F} \sum_{f_2=1}^{F} \boldsymbol{X}_{nf_1}^Q \boldsymbol{W}_{f_1 d}^Q \boldsymbol{W}_{f_2 d}^K \boldsymbol{X}_{mf_2}^K. \tag{B.2c}$$

Denote be the result of the subset of nodes in the softmax box as $\boldsymbol{A}$.

$$\overline{\boldsymbol{A}}_{nm} = \sum_{d=1}^{D} \sum_{f_1=1}^{F} \sum_{f_2=1}^{F} \overline{\boldsymbol{X}}_{nf_1}^Q \overline{\boldsymbol{W}}_{f_1 d}^Q \overline{\boldsymbol{W}}_{f_2 d}^K \overline{\boldsymbol{X}}_{mf_2}^K. \tag{B.3}$$

Comparing Equations (B.2c) and (B.3), we have $\boldsymbol{A} = \overline{\boldsymbol{A}}$. Let $\boldsymbol{B} = \text{softmax}(\boldsymbol{A})$ and $\overline{\boldsymbol{B}} = \text{softmax}(\overline{\boldsymbol{A}})$, we further have $\boldsymbol{B} = \text{softmax}(\boldsymbol{A}) = \text{softmax}(\overline{\boldsymbol{A}}) = \overline{\boldsymbol{B}}$.

Let $\overline{\boldsymbol{M}}$ be the result of the tensor diagram and $\boldsymbol{M}$ be the matrix representation of the single-head attention. We have

$$\boldsymbol{M}_{nd} = \sum_{m=1}^{N} \sum_{f=1}^{F} \boldsymbol{B}_{nm} \boldsymbol{X}_{mf}^{V} \boldsymbol{W}_{fd}^{V}. \tag{B.4}$$

$$\overline{\boldsymbol{M}}_{nd} = \sum_{m=1}^{N} \sum_{f=1}^{F} \overline{\boldsymbol{B}}_{nm} \overline{\boldsymbol{X}}_{mf}^{V} \overline{\boldsymbol{W}}_{fd}^{V}. \tag{B.5}$$

Comparing Equations (B.4) and (B.5), we have $\boldsymbol{C}$ is equivalent to $\overline{\boldsymbol{C}}$, which completes the proof. $\square$

## B.2 MULTI-HEAD SELF-ATTENTION (MHSA)

We now prove that tensor diagram in Figure 9c is equivalent to Equation (1c).

*Proof.* Let $H$ be the number of heads. Let $\overline{\boldsymbol{\mathcal{A}}} \in \mathbb{R}^{H \times N \times M}$ be the result of a contraction of the nodes within the softmax box. From the last section, we know that $\overline{\boldsymbol{\mathcal{A}}}_{h,:,:} = \boldsymbol{A}_{[h]} = \boldsymbol{Q}_{[h]} \boldsymbol{K}_{[h]}^{\top}$. Applying the softmax function $\overline{\boldsymbol{\mathcal{B}}} = \text{softmax}(\overline{\boldsymbol{\mathcal{A}}})$, we have $\overline{\boldsymbol{\mathcal{B}}}_{h,:,:} = \text{softmax}(\boldsymbol{Q}_{[h]} \boldsymbol{K}_{[h]}^{\top})$.

Let $\overline{\boldsymbol{T}}$ be result of the tensor diagram and $\boldsymbol{T}$ be that of Equation (B.1c).

$$\boldsymbol{T}_{nf_4} = \sum_{m,h,d_1,d_2} \text{softmax}(\boldsymbol{Q}_{[h],nd_1} \boldsymbol{Q}_{[h],md_1}) \boldsymbol{V}_{[h],md_2} \boldsymbol{W}_{[h],d_2 f_4}^{O}. \tag{B.6}$$

$$\overline{\boldsymbol{T}}_{n1f_4} = \sum_{m,h,d_1,d_2} \text{softmax}(\overline{\boldsymbol{\mathcal{Q}}}_{hnd_1} \overline{\boldsymbol{\mathcal{K}}}_{hmd_1}) \overline{\boldsymbol{\mathcal{V}}}_{hmd_2} \overline{\boldsymbol{\mathcal{W}}}_{hmd_2}^{O}. \tag{B.7}$$

Comparing $\overline{\boldsymbol{T}}$ and $\boldsymbol{T}$, we complete the proof.

$\square$

## C CLOSER INVESTIGATION OF MHSA AND ITS COMPARISON TO THSA

**Rethinking MHSA.** There are two types contractions in MHSA as shown in Figure 4a: *Head-Contraction*, a global contraction among all weights $\boldsymbol{\mathcal{W}}^{Q}, \boldsymbol{\mathcal{W}}^{K}, \boldsymbol{\mathcal{W}}^{V}$ and $\boldsymbol{\mathcal{W}}^{O}$ along mode $H$, and *Latent-Contraction*, a local contraction between $\boldsymbol{\mathcal{W}}^{Q}$ and $\boldsymbol{\mathcal{W}}^{K}$ as well as between $\boldsymbol{\mathcal{W}}^{V}$ and $\boldsymbol{\mathcal{W}}^{O}$ both along mode $D$. If we remove the Head-Contraction and maintain the Latent-Contraction only, multi-head is reduced to single-head with an additional contraction with the weight matrix $\boldsymbol{\mathcal{W}}^{O}$. To understand the role of the two types of contractions, we propose ablation studies by removing one of them and study the changes in the structure and the expressive power. It turns out that in both situation, *MHSA falls into a special case of THSA*.

**Repeated Proposition 1 with two special cases.** *Given $R = DH$ and other hyperparameters being the same, THSA module includes MHSA as a special case. Specifically,*
*(1) If $\boldsymbol{C} = \boldsymbol{1}_D \boldsymbol{1}_D^{\top}$, i.e., an $D \times D$ all-ones matrix, the THSA module reduces to an MHSA with a single head and latent dimension $D$ (i.e., a single-head self-attention with latent dimension $D$).*
*(2) If $\boldsymbol{C} = \boldsymbol{I}_H$, i.e., an $H \times H$ identity matrix, the THSA reduces to a MHSA with $H$ heads and latent dimension 1 (i.e., a heads-only self-attention with $H$ heads).*
*(3) If $\boldsymbol{C} = \boldsymbol{I}_H \otimes (\boldsymbol{1}_D \boldsymbol{1}_D^{\top})$, i.e., a Kronecker product of an all-ones matrix $\boldsymbol{1}_D \boldsymbol{1}_D^{\top} \in \mathbb{R}^{D \times D}$ and an identity matrix $\boldsymbol{I}_H \in \mathbb{R}^{H \times H}$, the THSA reduces to an MHSA with $H$ heads and latent dimension $D$.*

The proof of the above propositions is given in Appendix D.

**Definition 2** (Stable Rank). *Let $\boldsymbol{A} \in \mathbb{R}^{N \times M}$, the stable rank of $\boldsymbol{A}$ of is defined as $\sum_i \sigma_i^2 / \max_i \sigma_i$ where $\sigma_i$ is the singular value of $\boldsymbol{A}$.*

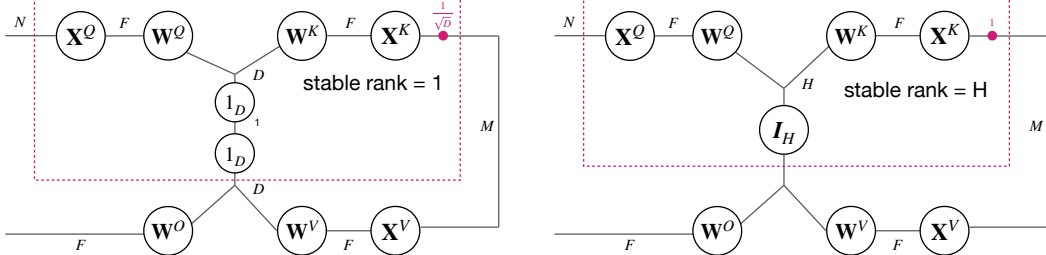

**Figure 10: (Left)** THSA includes a single-head self-attention. In this case, the stable rank of the core tensor $C$ is 1. It is interesting to observe that this is a single-head self-attention and this structure can be obtained by removing the head contraction from Figure 4a. **(Right)** THSA includes a MHSA with $H$ heads and latent dimension $= 1$. In this case, the stable rank equals to the rank of the identical matrix, which is $H$, corresponding to the number of heads of MHSA. Also, the structure on the right can be obtained by removing the latent contraction edges $D$ from Figure 4a.

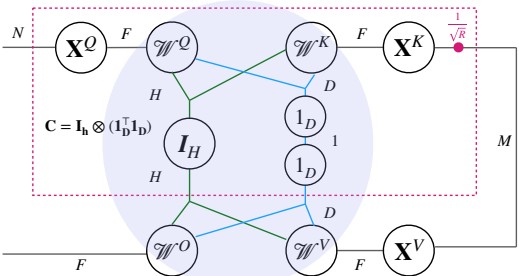

**Figure 11:** This figure shows how to initialize THSA with MHSA. MHSA is a special case of THSA when the core tensor $C$ takes the Kronecker product of a $D \times D$ all-one matrix and a $H \times H$ identity matrix. Note that in this case, the number of heads $H$ in MHSA is equivalent to the stable rank of $C$.

**The notion of heads in THSA.** The concept of the number of heads in THSA generalizes to the stable rank of the core matrix, allowing a data-driven implicit training. The number of heads $H$ in MHSA corresponds to the *stable rank* of the core matrix $C$, defined as $\sum_i \sigma_i^2 / \max_i \sigma_i$ where $\sigma_i$ is the singular value. It is a useful surrogate for the rank because it is largely unaffected by tiny singular values.

As shown in Appendix C when $C$ is constructed by the outer product of two vectors, the stable rank of $C$ is 1, while in this case, the whole structure is equivalent to a single-head self-attention. When $C$ is an identical matrix as shown in appendix C, the stable rank of $C$ equals to the rank of the identical matrix, which is $H$, corresponding to the number of heads of MHSA. Also, the structure on the right can be obtained by removing the latent contraction edges $D$ from Figure 4a. When $C$ is the Kronecker product of the all-one matrix and the identical matrix, which corresponds to MHSA, the stable rank of $C$ is $H$, and this number does not change in the training because that $C$ is a fixed matrix in MHSA.

# D  SUPPLEMENTARY MATERIAL FOR TUFORMER AND PROOFS

We first formally describe how to compare the expressive power between two models.

**Definition 3** (Expressive Power)**.** *Suppose we have two function classes* $\mathbb{F}$, $\mathbb{G}$ *with the same source domain* $\mathcal{X}$ *and target domain* $\mathcal{Y}$, *i.e., each function* $f \in \mathbb{F}$ *(or* $g \in \mathbb{G}$*) is a mapping from* $\mathcal{X}$ *to* $\mathcal{Y}$*. We say* $\mathbb{G}$ *is more expressive than* $\mathbb{F}$ *if* $G \supseteq F$*: for any* $f \in \mathcal{F}$*, there exists* $g \in G$ *such that* $g(x) = f(x), \forall x \in \mathcal{X}$*. Furthermore, we say* $\mathbb{G}$ *is strictly more expressive than* $\mathbb{F}$ *if* $\mathbb{G} \supset \mathbb{F}$*: besides* $G \supseteq F$*, there exists* $g \in \mathbb{G}$ *such that given any* $f$ *there exists* $x \in \mathcal{X}$ *and* $g(x) \neq f(x)$*.*

According to the definition, expressive power is a partial order set. Therefore, expressive power of different models is usually not comparable as two sets might have overlap and non-overlap regions. However, when one is a superset or subset of the other, we can unambiguously compare their expressive power. In the following theorem, we formally show that a Tunable-head self-attention (THSA) module has a higher expressive than multi-head self-attention (MHSA) if the rank $R$ in THSA is equal to the product of the number of heads $H$ and the latent dimension $D$ in MHSA.

**Theorem 4** (THSA is more expressive than MHSA). *Given $R = DH$ and other hyper-parameters being the same, a Tunable-head self-attention (THSA) module with rank $R$ is more expressive than a multi-head self-attention (MHSA) module with $H$ heads and latent dimension $D$.*

Theorem 4, proved in Appendix D.2, shows that any function realizable by MHSA can be realized by THSA.

In this section, we will prove Proposition 1 and Theorem 4 in Section 3. In both theorems, we assume the rank in *Tunable-head self-attention* (THSA) equals to the product of the number of heads $H$ and the latent dimension $D$ in *multi-head self-attention* (MHSA), i.e., $R = DH$.

For convenience of reference, we recap the mathematical expressions and tensor diagrams for both multi-head self-attention (MHSA) and Tunable-head self-attention (THSA) in the following.

**Multi-head Self-Attention (MHSA).** An MHSA module has its learnable parameters as $H$ sets of (query, key, value) weight matrices $\{\boldsymbol{W}_{[h]}^Q, \boldsymbol{W}_{[h]}^K, \boldsymbol{W}_{[h]}^V\}_{h=1}^H$ (with $\boldsymbol{W}_{[h]}^Q, \boldsymbol{W}_{[h]}^K, \boldsymbol{W}_{[h]}^V \in \mathbb{R}^{F \times D}$ for each $h$) and an output weight matrix $\boldsymbol{W}^O \in \mathbb{R}^{HD \times F}$. The module maps three input matrices $\boldsymbol{X}^Q \in \mathbb{R}^{N \times F}, \boldsymbol{X}^K, \boldsymbol{X}^V \in \mathbb{R}^{M \times F}$ into an output matrix $\boldsymbol{M} \in \mathbb{R}^{N \times F}$.

$$\boldsymbol{Q}_{[h]} = \boldsymbol{X}^Q \boldsymbol{W}_{[h]}^Q;\ \boldsymbol{K}_{[h]} = \boldsymbol{X}^K \boldsymbol{W}_{[h]}^K;\ \boldsymbol{V}_{[h]} = \boldsymbol{X}^V \boldsymbol{W}_{[h]}^V, \tag{D.1a}$$

$$\textbf{head}_{[h]} = \text{softmax}\left(\boldsymbol{Q}_{[h]} \boldsymbol{K}_{[h]}^\top\right) \boldsymbol{V}_{[h]}, \tag{D.1b}$$

$$\boldsymbol{M} = \left[\textbf{head}_{[1]},\ \textbf{head}_{[2]},\ \cdots,\ \textbf{head}_{[H]}\right] \boldsymbol{W}^O. \tag{D.1c}$$

In Equation (D.1a), $\boldsymbol{Q}_{[h]} \in \mathbb{R}^{N \times D}, \boldsymbol{K}_{[h]}, \boldsymbol{V}_{[h]} \in \mathbb{R}^{M \times D}$ are query, key, value matrices respectively. In Equation (D.1b), we omit the scaling factor $1/\sqrt{D}$ for the softmax function (the scalar can be merged into the learnable parameters), and $\textbf{head}_{[h]} \in \mathbb{R}^{N \times D}$ is the resulted matrix for head $h$.

**Tunable-Head Self-Attention.** A THSA module is parameterized by five matrices: a query weight matrix $\overline{\boldsymbol{W}}^Q \in \mathbb{R}^{F \times R}$, a key weight matrix $\overline{\boldsymbol{W}}^K \in \mathbb{R}^{F \times R}$, a value weight matrix $\overline{\boldsymbol{W}}^V \in \mathbb{R}^{F \times R}$, an output weight matrix $\overline{\boldsymbol{W}}^O \in \mathbb{R}^{R \times F}$, and an additional core matrix $\boldsymbol{C} \in \mathbb{R}^{R \times R}$. The module has the same input and output domains as in MHSA, i.e., it takes three matrices $\boldsymbol{X}^Q \in \mathbb{R}^{N \times F}$, $\boldsymbol{X}^K, \boldsymbol{X}^V \in \mathbb{R}^{M \times F}$ as inputs and returns a matrix $\boldsymbol{T} \in \mathbb{R}^{N \times F}$ as output.

$$\overline{\boldsymbol{Q}} = \boldsymbol{X}^Q \overline{\boldsymbol{W}}^Q;\ \overline{\boldsymbol{K}} = \boldsymbol{X}^K \overline{\boldsymbol{W}}^K;\ \overline{\boldsymbol{V}} = \boldsymbol{X}^V \overline{\boldsymbol{W}}^V, \tag{D.2a}$$

$$\overline{\textbf{head}}_r = \text{softmax}\left(\sum_{s=1}^R \boldsymbol{C}_{rs}\, \overline{\boldsymbol{Q}}_s \overline{\boldsymbol{K}}_s^\top\right) \overline{\boldsymbol{V}}_r, \tag{D.2b}$$

$$\boldsymbol{T} = \overline{\textbf{head}}\, \overline{\boldsymbol{W}}^O = \left[\overline{\textbf{head}}_1, \overline{\textbf{head}}_2, \cdots, \overline{\textbf{head}}_R\right]. \tag{D.2c}$$

To distinguish THSA from MHSA, we use over-scored symbols whenever needed. In Equation (D.2a), $\overline{\boldsymbol{Q}} \in \mathbb{R}^{N \times R}, \overline{\boldsymbol{K}}, \overline{\boldsymbol{V}} \in \mathbb{R}^{M \times R}$ are query, key, value matrices respectively. In Equation (D.2b), we again omit the scaling factor $1/\sqrt{R}$ in the softmax function. Equation (D.2b) leads to a head matrix $\overline{\textbf{head}} \in \mathbb{R}^{F \times N}$, where $\overline{\textbf{head}}_r \in \mathbb{R}^F$ denotes the $r^{\text{th}}$ column of the matrix $\overline{\textbf{head}}$.

### D.1 PROOF OF PROPOSITION 1

*Proof of Proposition 1.* We constructively prove that THSA reduces to three cases of MHSA if the query/key/value weight matrices in THSA concatenate the corresponding matrices in MHSA:

$$\overline{\boldsymbol{W}}^Q = \left[\boldsymbol{W}_{[1]}^Q, \boldsymbol{W}_{[2]}^Q, \cdots, \boldsymbol{W}_{[H]}^Q\right], \tag{D.3a}$$

$$\overline{\boldsymbol{W}}^K = \left[\boldsymbol{W}_{[1]}^K, \boldsymbol{W}_{[2]}^K, \cdots, \boldsymbol{W}_{[H]}^K\right], \tag{D.3b}$$

$$\overline{\boldsymbol{W}}^V = \left[\boldsymbol{W}_{[1]}^V, \boldsymbol{W}_{[2]}^V, \cdots, \boldsymbol{W}_{[H]}^V\right]. \tag{D.3c}$$

In addition, we set $\overline{\boldsymbol{W}}^O = \boldsymbol{W}^O$. We have $\overline{\boldsymbol{W}}^Q, \overline{\boldsymbol{W}}^K, \overline{\boldsymbol{W}}^V \in R^{F \times R}$, and $\overline{\boldsymbol{W}}^O \in \mathbb{R}^{R \times F}$.

**(1)** For $\boldsymbol{C} = \boldsymbol{1}_R \boldsymbol{1}_R^\top$, i.e. an $R \times R$ all-ones matrix, we aim to prove that the THSA module reduces to a single-head self-attention with $H = 1$ and $D = R$. Since there is only one head in the module, we have $\overline{\boldsymbol{W}}^Q = \boldsymbol{W}_{[1]}^Q, \overline{\boldsymbol{W}}^K = \boldsymbol{W}_{[1]}^K$, and $\overline{\boldsymbol{W}}^V = \boldsymbol{W}_{[1]}^V$. As an immediate result,

$$\overline{\boldsymbol{Q}} = \boldsymbol{X}^Q \overline{\boldsymbol{W}}^Q = \boldsymbol{X}^Q \boldsymbol{W}_{[1]}^Q = \boldsymbol{Q}_{[1]}, \tag{D.4a}$$

$$\overline{\boldsymbol{K}} = \boldsymbol{X}^K \overline{\boldsymbol{W}}^K = \boldsymbol{X}^K \boldsymbol{W}_{[1]}^K = \boldsymbol{K}_{[1]}, \tag{D.4b}$$

$$\overline{\boldsymbol{V}} = \boldsymbol{X}^V \overline{\boldsymbol{W}}^V = \boldsymbol{X}^V \boldsymbol{W}_{[1]}^V = \boldsymbol{V}_{[1]}. \tag{D.4c}$$

Since $\boldsymbol{C}$ is an all-ones matrix, i.e., $\boldsymbol{C}_{rs} = 1, \forall r, s$, we rewrite Equation (D.2b) as:

$$\overline{\textbf{head}}_r = \text{softmax}\left(\sum_{s=1}^R \overline{\boldsymbol{Q}}_s \overline{\boldsymbol{K}}_s^\top\right)\overline{\boldsymbol{V}}_r = \text{softmax}\left(\overline{\boldsymbol{Q}}\,\overline{\boldsymbol{K}}^\top\right)\overline{\boldsymbol{V}}_r. \tag{D.5}$$

Notice that the equation holds for each column $r$, we further write all $R$ equations jointly as:

$$\overline{\textbf{head}} = \text{softmax}\left(\overline{\boldsymbol{Q}}\,\overline{\boldsymbol{K}}^\top\right)\overline{\boldsymbol{V}} = \text{softmax}\left(\boldsymbol{Q}_{[1]}\boldsymbol{K}_{[1]}^\top\right)\boldsymbol{V}_{[1]} = \textbf{head}_{[1]}. \tag{D.6}$$

Finally, we express the output matrix as:

$$\boldsymbol{T} = \overline{\textbf{head}}\,\overline{\boldsymbol{W}}^O = \textbf{head}_{[1]}\boldsymbol{W}^O = \boldsymbol{M}, \tag{D.7}$$

which concludes the reduction for the special case **(1)**.

**(2)** For $\boldsymbol{C} = \boldsymbol{I}_R$, an $R \times R$ identity matrix, we aim to prove that the THSA module reduces to a heads-only self-attention with $H = R$ and $D = 1$. Since the latent dimension $D = 1$, i.e., each $\boldsymbol{W}_{[h]}^Q$ (or $\boldsymbol{W}_{[h]}^K$, $\boldsymbol{W}_{[h]}^V$) is a vector, we have $\overline{\boldsymbol{W}}_r^Q = \boldsymbol{W}_{[r]}^Q, \overline{\boldsymbol{W}}_r^K = \boldsymbol{W}_{[r]}^K$, and $\overline{\boldsymbol{W}}_r^V = \boldsymbol{W}_{[r]}^V$. Therefore,

$$\overline{\boldsymbol{Q}}_r = \boldsymbol{X}^Q \overline{\boldsymbol{W}}_r^Q = \boldsymbol{X}^Q \boldsymbol{W}_{[r]}^Q = \boldsymbol{Q}_{[r]}, \tag{D.8a}$$

$$\overline{\boldsymbol{K}}_r = \boldsymbol{X}^K \overline{\boldsymbol{W}}_r^K = \boldsymbol{X}^K \boldsymbol{W}_{[r]}^K = \boldsymbol{K}_{[r]}, \tag{D.8b}$$

$$\overline{\boldsymbol{V}}_r = \boldsymbol{X}^V \overline{\boldsymbol{W}}_r^V = \boldsymbol{X}^V \boldsymbol{W}_{[r]}^V = \boldsymbol{V}_{[r]}. \tag{D.8c}$$

Since $\boldsymbol{C}$ is an identity matrix, i.e., $\boldsymbol{C}_{rr} = 1$ and $\boldsymbol{C}_{rs} = 0, \forall r \neq s$, we rewrite Equation (D.2b) as:

$$\overline{\textbf{head}}_r = \text{softmax}\left(\overline{\boldsymbol{Q}}_r \overline{\boldsymbol{K}}_r^\top\right)\overline{\boldsymbol{V}}_r = \text{softmax}\left(\boldsymbol{Q}_{[r]}\boldsymbol{K}_{[r]}^\top\right)\boldsymbol{V}_{[r]} = \textbf{head}_{[r]}. \tag{D.9}$$

Finally, we express the output matrix as:

$$\boldsymbol{T} = \overline{\textbf{head}}\,\overline{\boldsymbol{W}}^O = \left[\textbf{head}_{[1]}, \textbf{head}_{[2]}, \cdots; \textbf{head}_{[R]}\right]\boldsymbol{W}^O = \boldsymbol{M}, \tag{D.10}$$

which concludes the reduction for the special case **(2)**.

**(3)** For $\overline{\boldsymbol{C}} = \boldsymbol{I}_H \otimes (\boldsymbol{1}_D \boldsymbol{1}_D^\top)$, i.e., a Kronecker product between an identity matrix $\boldsymbol{I}_H \in \mathbb{R}^{H \times H}$ and an all-ones matrix $\boldsymbol{1}_D \boldsymbol{1}_D^\top \in \mathbb{R}^{D \times D}$, we aim to prove that the THSA module reduces to a multi-head

self-attention (MHSA) module with $H$ heads and latent dimension $D$. In this general case, we have

$$\overline{Q} = X^Q \overline{W}^Q = X^Q \left[ W_{[1]}^Q, \ W_{[2]}^Q, \ \cdots, \ W_{[H]}^Q \right] = \left[ Q_{[1]}, \ Q_{[2]}, \ \cdots, \ Q_{[H]} \right] \tag{D.11}$$

$$\overline{K} = X^K \overline{W}^K = X^K \left[ W_{[1]}^K, \ W_{[2]}^K, \ \cdots, \ W_{[H]}^K \right] = \left[ K_{[1]}, \ K_{[2]}, \ \cdots, \ K_{[H]} \right] \tag{D.12}$$

$$\overline{V} = X^V \overline{W}^V = X^V \left[ W_{[1]}^V, \ W_{[2]}^V, \ \cdots, \ W_{[H]}^V \right] = \left[ V_{[1]}, \ V_{[2]}, \ \cdots, \ V_{[H]} \right] \tag{D.13}$$

Notice that $C$ is a block diagonal matrix $\mathsf{blkdiag}(\mathbf{1}_D \mathbf{1}_D^\top, \cdots, \mathbf{1}_D \mathbf{1}_D^\top)$, where each block is an all-ones matrix. Therefore we can rewrite Equation (D.2b) for $r = (h-1)D + d$ as:

$$\overline{\mathsf{head}}_{(h-1)D+d} = \mathsf{softmax} \left( \sum_{s=1}^R \mathbb{1}_{(h-1)D < s \leq hD} \left( \overline{Q}_s \overline{K}_s^\top \right) \right) \overline{V}_r \tag{D.14}$$

$$= \mathsf{softmax} \left( \sum_{d=1}^D Q_{[h],d} K_{[h],d}^\top \right) V_{[h],d} \tag{D.15}$$

$$= \mathsf{softmax} \left( Q_{[h]} K_{[h]}^\top \right) V_{[h],d}. \tag{D.16}$$

Since Equation (D.16) holds for each column $d$, we write all $D$ equations jointly as:

$$\left[ \overline{\mathsf{head}}_{(h-1)D+d}, \ \cdots, \ \overline{\mathsf{head}}_{hD} \right] = \mathsf{softmax} \left( Q_{[h]} K_{[h]}^\top \right) V_{[h]} = \mathsf{head}_{[h]}. \tag{D.17}$$

Again, since Equation (D.17) holds for each block $h$, we combine all $H$ equations as:

$$\overline{\mathsf{head}} = \left[ \mathsf{head}_{[1]}, \ \mathsf{head}_{[2]}, \ \cdots, \ \mathsf{head}_{[H]} \right]. \tag{D.18}$$

Finally, we express the output matrix as:

$$T = \overline{\mathsf{head}} \ \overline{W}^O = \left[ \mathsf{head}_{[1]}, \ \mathsf{head}_{[2]}, \ \cdots, \ \mathsf{head}_{[H]} \right] W^O = M, \tag{D.19}$$

which concludes the reduction for the case **(3)**. $\qquad\square$

### D.2  Proof of Theorem 4

*Proof of Theorem 4.* We have shown that THSA reduces to MHSA when $C$ takes specific forms. This shows that THSA is more expressive than MHSA — any mapping by an MHSA module can also be realized by a THSA module.

To prove that THSA is *strictly* more expressive than MHSA, we need find a mapping by a THSA module that can not be realized by an MHSA module. It suffices to show that the matrix $M$ is rank-deficient, while the matrix $T$ can be full-rank. We assume $MH < \min(N, F)$.

For convenience, we divide the matrix $W^O$ for MHSA into $H$ blocks.

$$W^O = \left[ W_{[1]}^{O \top}, \ W_{[2]}^{O \top}, \ \cdots, \ W_{[H]}^{O} \right]^\top, \tag{D.20}$$

where $W_{[h]}^O \in \mathbb{R}^{D \times F}$ is the matrix for the $h^{\text{th}}$ head. We now rewrite the output matrix $M$ as:

$$M = \sum_{h=1}^H \underbrace{\mathsf{softmax} \left( Q_{[h]} K_{[h]}^\top \right)}_{\mathbb{R}^{N \times M}} \underbrace{\left( V_{[h]} W_{[h]}^O \right)}_{\mathbb{R}^{M \times F}}. \tag{D.21}$$

Since the matrix inside summation is a product of two matrices of size $\mathbb{R}^{N \times M}$ and $\mathbb{R}^{M \times F}$, its rank is at most $M$. Use the property $\mathsf{rank}(A + B) \leq \mathsf{rank}(A) + \mathsf{rank}(B)$, we have $\mathsf{rank}(M) \leq MH < \min(N, F)$. Similarly, we divide the matrix $\overline{W}^O$ for THSA into $R$ rows and rewrite the output matrix $T$ as:

$$T = \sum_{r=1}^R \underbrace{\mathsf{softmax} \left( \sum_{s=1}^R C_{rs} \overline{Q}_{:,r} \overline{K}_{:,r}^\top \right)}_{\mathbb{R}^{N \times M}} \underbrace{\left( \overline{V}_{:,r} \overline{W}_{r,:}^O \right)}_{\mathbb{R}^{M \times F}}. \tag{D.22}$$

Using the same argument, we have $\mathsf{rank}(T) \leq MR$. Since $H < R$ (in fact, $R = DH$), we can always find an example such that $\mathsf{rank}(T) > \mathsf{rank}(M)$. This completes the proof. $\qquad\square$

# E  SUPPLEMENTARY MATERIAL FOR EXPERIMENTS

All the experiments are run on computing nodes with 4 NVIDIA GeForce GPUs.

**(1) Language Modeling on Penn Treebank.** Language modeling is the task of computing the probability of a sentence or a sequence of words. The model performance is measured by *per-word Perplexity (PPL)* which is the lower the better.

Penn Treebank (Marcus et al., 1993) is under the LDC User Agreement for Non-Members. We adopt the default settings as in the vanilla Transformer models(Vaswani et al., 2017). Specifically, we have $N$ (length of the sequence) to be 2048, $F$ (length of the initial embedding) to be 512, 8 number of heads, 6 layers encoder-decoder structure, $D$ (latent feature dimension) to be 64. We use Adam optimizer and have 4000 warm-up steps. We also use label smoothing(0.1). The model is trained for 50 epochs.

**(2) Neural Machine Translation on WMT16.** The goal of NMT is to generate a corresponding sequence in one language given a sequence in another. The performance is measured by *BLEU* scores. (the higher, the better).

WMT16 English-German dataset (Sennrich et al., 2016) is under the MIT license. In this experiment, we use a Transformer model with 8 layers and we have $F = 512$ and $R = 1024$. The learning rate is $1e^{-5}$ and 4000 warmups. We also set label-smooting to be $0.1$. To speed up the experiment, we use the mix-precision trick with an optimization level being O1.

**(3) Image Generation on MNIST and CIFAR-10.** The image generation task is to predict an image pixel by pixel. The performance is evaluated by *bits per dimension (BPD)*. The image generation task is chosen to demonstrate, firstly, that Tuformers work beyond the language domain, and secondly, Tuformers can improve other efficient designs.

CIFAR10 (Krizhevsky, 2009) dataset is under the MIT license (MIT), MNIST(LeCun et al., 2010) is under the Creative Commons Attribution-Share Alike 3.0 license. The reason why we choose to evaluate Tuformer in image generation tasks on these two datasets is that we want to show the compatibility of our model to other efficient designs. Katharopoulos et al. (2020) introduces a linear transformer that has a state-of-the-art complexity. As a result, we evaluate our design on the same experiments as they do in their paper. We adopt basically the same settings except for GPU resources. Concretely, we use a 8 layer transformer model with 8 heads. The embedding size $F$ is 256 an $D$ is 32. $N$ is set to be 1024. We also use "RAdam" (Ma & Yarats, 2021) optimizer for a state-of-the-art result. The model is trained for 250 epochs for MNIST dataset and 50 epochs for CIFAR10 dataset. The batch size is 10 in MNIST training and 4 in CIFAR training. We additionally use he mix-precision trick with an optimization level set to be O1 to speed up the training.

**(4) Automatic Speech Recognition on LibriSpeech Dataset** ASR consists of transcribing audio speech segments into text. The performance is measured by *Word Error Rate (WER)*. (the lower, the better). We adopt the default settings as in Fairseq (Ott et al., 2019), except that we replace MHSA with THSA.

**(5) Natural Language Inference on MNLI and QNLI.** NLI is a task of determining whether the given "hypothesis" and "premise" logically follow (entailment) or unfollow (contradiction) or are undetermined (neutral) to each other. The results are evaluated by accuracy. Using MNLI dataset, the model tries to predict whether sentence A entails or contradicts B while in QNLI dataset, the model is trained to answer whether sentence B contains answers to the question in sentence A. We use a batch size of 32 for 10 epochs, with a learning rate $1e^{-5}$.

**Memory Overhead in THSA.** As mentioned in Section 4.3, naive Tuformers come with an extra memory overhead when calculating values within the softmax. In Tuformers, if done naively, the memory overhead is $MNR$ where $N, M$ denote the sequence length and $R = DH$ is the rank of the core tensor, incurring a $D$ times larger memory overhead than vanilla Transformers in calculating softmax. We argue that combining Tuformers with kernel-based efficient variants such as Linear Transformers (Katharopoulos et al., 2020) and polynomial kernels (Tsai et al., 2019) kernels could not only cancel the memory overhead but also achieve the same linear memory complexity by removing the nonlinear constrain in the softmax.

Furthermore, the extra memory overhead can be eliminated from the engineering side using the check-pointing technique: intermediate values are not saved in the memory but recalculated when needed. We compare the runtime in both training and inference phase with check-pointing. As shown in fig. 12, under the same number of parameters, there is only slight increase in the training and basically the same runtime in the inference phase. With a direct transformation from Transformer to Tuformer (for example, when initialized with pre-trained models), the runtime is still under a reasonable range.

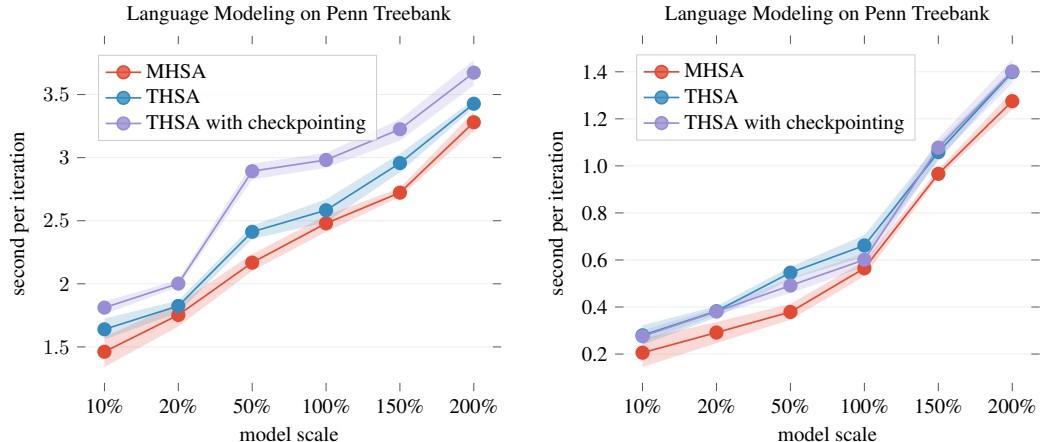

**Figure 12:** **(Left)** Training and **(Right)** inference run-time comparison between MHSA, THSA with the same number of parameters and THSA with the same number of parameters with check-pointing. The batch size is 256.

