# OpenReview forum: "Tuformer: Data-driven Design of Transformers for Improved Generalization or Efficiency"
_ICLR.cc/2022/Conference — ICLR 2022 Poster_

### Official Review · Reviewer_hfb5 · 2021-11-01

**Correctness:** 3
**Technical Novelty And Significance:** 3
**Empirical Novelty And Significance:** 3
**Recommendation:** 5
**Confidence:** 4

**Main Review:**

Tensor network diagram seems unnecessary to the paper. Using Eq. (2a,3b,2c) and Eq. (1a,1b,1c) is enough for readers to understand the difference between MHSA and THSA. The diagrams in figure 4 convey the same idea but involve more technical content. The first contribution claimed by the author could be safely erased without affecting the completeness of this work.  Let me know if I am wrong.

In Sec. 3.2, as a reader, I am more curious about the proof for theorem 3 than definition 2. But the authors put definition 2 in the main paper and proof for theorem 3 in the appendix -- why not do the opposite?


The author claimed "THSA *significantly* improves MHSA", I am wondering whether the author actually did the significant test (as I encourage the authors to do it), or it would better use another word like "notably" instead.


Please consider *quantitatively* showing the inference time.

Subscripts in Figure (3) seem wrong, please have a check. A_mr B_nr not  A_mr B_nr. The same happens in the text related to Eq 2, please double-check the superscripts there (the shape of X^Q, W^Q, Q, Q_r).


The technical content is quite close to  Cordonnier et.al 2020 https://openreview.net/forum?id=bK-rJMKrOsm , please consider explaining the difference. See figure 2  (left) in Cordonnier et.al 2020, the standard SAN can be considered as a special case of theirs. This is quite similar to  Proposition 1 in this paper -- of course, they are not the same, but with some differences.





**Summary Of The Paper:**


The paper proposes a THSA that generalizes the standard self-attention. THSA empirically outperforms the standard self-attention.

**Summary Of The Review:**

The paper introduces a general Self-attention, which is interesting and insightful. However, a general model that costs more (time or space) but slightly performs better is not that attractive.

---

> ### Author Response · Authors · 2021-11-16
> **To Reviewer hfb5: Importance of tensor diagram and comparison with Cordonnier's paper.**
>
> We thank Reviewer hfb5 for the effort in reviewing the paper. We hope our response could raise the Reviewer's attention to our contribution of introducing tensor diagram to the ML community. Tensor diagram could not only serve as an analytical visualization tool for comparing expressive powers between different architectures, but also could spark interest and exploration on how to design neural network architecture using tensor representation theory.
>
> > Q1 The first contribution claimed by the author could be safely erased without affecting the completeness of this work.
>
> The tensor diagram is nothing but an intuitive graphical representation of rigorous math formulas. As a result, we can always mathematically formulate and complete any idea inspired by tensor diagram representations.
>
> However, the development of Tuformer is inspired by the observation that the MHSA is a special case of the proposed THSA where the core matrix $\mathbf{C}$ is constant. Without tensor diagram, this will be very difficult to achieve whereas this is intuitive with tensor diagram. To be concrete, $\mathbf{C}$ is the Kronecker product of **(a)** the outer product of two all-one vectors of dimension $D$ and **(b)** an identity matrix of dimension $H \times H$ (Proposition 1).  Then, it is natural to consider making $\mathbf{C}$ tunable and inspecting the expressive power of the new structure.
>
> Moreover, comparing the expressive power of neural networks is equivalent to comparing tensor structures in the tensor diagram, which could spark interest and exploration on how to design neural network architecture using tensor representation theory.
>
> Therefore, although, mathematically, a tensor diagram is not required to prove the theorem, it is indispensable and valuable to this paper and also worth being introduced to the ML community.
>
> > Q2  The author claimed "THSA significantly improves MHSA", I am wondering whether the author actually did the significant test (as I encourage the authors to do it), or it would better use another word like "notably" instead.
>
> We repeated each train-from-scratch experiment multiple times and they are denoted as shadow areas in the figures. For those with pre-trained models, we will do that and provide the information once we have them.
>
> > Q3 Difference between Tuformer and Cordonnier's work
>
> **1. Difference in Motivation.**
> - The collaborative attention is a compression model. It starts from the empirical evidence that different pairs of weight matrices $\mathbf{W}^Q$, $\mathbf{W}^K$ attend to similar subspaces. Therefore, they proposed a weight-sharing structure to improve the parameter efficiency of the proposed model.
> - This papers representd MHSA in tensor diagram and proposed a THSA with a guaranteed higher expressive power (including MHSA as a special case).
>
> **2. Difference in proposed architecture from a tensor perspective.**
> - The collaborative attention is a specific design in the proposed design space in this paper. It focuses on the relationship of $\mathbf{W}^Q$, $\mathbf{W}^K$ across heads and uses CP decomposition to learn $\mathbf{W}^Q$, $\mathbf{W}^K$ and a mixing matrix $\mathbf{M}$. The rest remains the same as MHSA.
> - Tuformer studies the nature of MHSA and discusses the relationship among the four weight matrices $\mathbf{W}^Q$, $\mathbf{W}^K$, $\mathbf{W}^V$ and $\mathbf{W}^O$, the core tensor in THSA is made trainable for a data-driven architecture.
>
> **3. Difference in contributions.**
>   - The collaborative attention is an empirical compression model and is parameter efficient.
>   - Our work formulates a design space and studies the expressive power of different architectures. The proposed THSA has a guaranteed higher expressive power. Also, the number of heads in THSA is trainable.
>   - This paper introduces tensor diagram representation to the ML community. We hope the study in this paper could spark more interest and inspire ideas for future designs of high-performance models with guarantees using tensor representations.
>
> > Q4 Notation issues.
>
> We thank Review hfb5 for pointing out the notation issues in the paper. We will make corresponding revisions and update it soon.
>
> > Q5 Inference Time
>
> We have attached the inference time in NMT and Language Modeling tasks for your reference. Full details will be updated in the revised version.
> In our current comparison, under the same model scale, Tuformer has 25% more parameters than MHSA. We will compare them under the same number of parameters and update the results once we have them.
>
> We thank Reviewer hfb5 for their valuable feedback. We have clarified all questions from Reviewer hfb5. Please don't hesitate to let us know if you have further questions. Thank you very much!

---

> > ### Author Response · Authors · 2021-11-16
> > **To Reviewer hfb5: inference task comparison in LM and NMT tasks.**
> >
> > Language Modeling
> >
> > | Model Scale | MHSA | THSA | THSA with ckpt |
> > |:----------- |:---- |:---- |:--------------:|
> > | 20%         | 0.32 | 0.37 |      0.37      |
> > | 50%         | 0.41 | 0.55 |      0.55      |
> > | 100%        | 0.54 | 0.67 |      0.68      |
> > | 150%        | 1.01 | 1.14 |      1.14      |
> > | 200%        | 1.27 | 1.57 |      1.57      |
> >
> > Neural Machine Translation
> >
> >
> > | Model Scale | MHSA | THSA | THSA with ckpt |
> > |:----------- |:---- |:---- |:--------------:|
> > | 20%         | 1.81 | 1.91 |      1.91      |
> > | 50%         | 2.21 | 2.70 |      2.71      |
> > | 100%        | 2.43 | 2.95 |      2.95      |
> > | 150%        | 3.81 | 4.14 |      4.14      |
> > | 200%        | 5.47 | 6.57 |      6.56      |

---

> > ### Comment · Reviewer_hfb5 · 2021-11-29
> > **Thanks for your replies**
> >
> > Thanks for your replies. I decide to keep my recommendation score unchanged.
> >
> > The general idea of this paper is interesting and novel. I also like the tensor diagram, which I believe is of great interest to the machine learning community, and I appreciate the authors' efforts. However, this does not mean this paper has the necessity to introduce it.
> >
> > I am not questioning the originality and novelty of this paper when mentioning Cordonnier's work. The difference in design may help readers to understand the contribution of this paper.
> >
> > It seems that Tuformer outperforms vanilla Transformer but becomes slightly slower and bigger (in terms of parameter numbers). This might make its deployment in real-world applications less helpful. Regarding its theoretical contribution, it does not seem significant from my point of view.

---

### Official Review · Reviewer_R5aP · 2021-11-01

**Correctness:** 3
**Technical Novelty And Significance:** 3
**Empirical Novelty And Significance:** 2
**Recommendation:** 6
**Confidence:** 4

**Main Review:**

The main idea of the paper, to interpret the weight tensors of MHSA as factors of a decomposition, is new and THSA appears to be supported by experimental results (although the results are so small that I wonder whether they are significant relative to hyper-parameter tuning and stochastic fluctuations of the training). However, no evidence is provided to justify the choice - why should it be Tucker Decomposition, and not Tensor Train or Tensor Ring decomposition? And how does that choice influence the properties of the attention mechanism?

The authors also claim the following contributions:
1. tensor diagram notation for attention mechanism
2. design space for attention block together with expressiveness analysis framework
3. initialisation scheme from pre-trained Transformer

While (3) has important applications in reusing pre-trained transformer models, (1) doesn't help much to understand the attention mechanism, as it will be readable only for people whose multi-linear algebra background is advanced enough to also reach the same understanding without diagrams. Whereas (2) effectively proves that, given some network, adding an additional parametric matrix somewhere in the middle of that network does not decrease its expressiveness, which is trivial

In my view, the general idea is valid and potentially interesting, but it would make the paper much stronger to revise it in terms of methodology and experiments, so as to actually study how different tensor decompositions influence the attention mechanism, and which ones might be suited best for specific layouts and tasks. However, this would be a completely different paper that would have to be reviewed again, so I am on the fence about the present version.

The presentation is mostly clear, but in a few places it is sloppy and appears unfinished and rushed. This would need to be reworked:
page 4 - duplicate paragraphs at the end of the page
page 5 - duplicate paragraphs in 3. and 3.1.
Definition 2: f \in X should be probably f \in F ?
(page 3: For instant -> For instance)

**Summary Of The Paper:**

The paper attempts for a better understanding of the mechanism of multi-head self-attention (MHSA). It proposes to interpret MHSA weight matrices as factors of the Tucker Decomposition and adds an additional core weight tensor to MHSA to obtain Tucker-head self-attention (THSA).

**Summary Of The Review:**

The paper address an interesting and relevant problem, namely generalisation of the MHSA mechanism, and it presents interesting, if somewhat terse and preliminary, experimental results. However, while the view as tensor decomposition is promising, the seemingly ad-hoc commitment to Tucker without a clear rationale or systematic study is unsatisfactory and leaves an unfinished impression. The paper does put forward a novel idea and has high potential, but it would require a major overhaul to fulfil that potential and turn it into a possibly really strong and useful contribution, rather than a fleeting quick-shot.

I find it difficult to give a recommendation for this paper: If one follows the conventional criteria, there is novelty and the potential to lead to interesting follow-up, so one might argue for acceptance, even though there isn't a strong case for future impact. On the other hand, it feels a bit premature and a bit more work could make it a lot better and more significant.

---

> ### Author Response · Authors · 2021-11-16
> **To Reviewer R5aP about the justification of the Tuformer structure and our contributions.**
>
> We appreciate Reviewer R6aP’s effort in reviewing the paper, and we will discuss how we justify the choice of the current structure. We hope that our response could clarify some wording-related concerns to the model structure and our contributions.
>
> In the manuscript, we justify the Tucker-head self-attention (THSA) in Section 3. Proposition 1 motivates the design, and Theorem 3 guarantees an increased expressive power of the Tuformer model.
>
> In this paper, we first represented MHSA in the tensor diagram. We observed that the MHSA is a special case of the proposed THSA where the core matrix $\mathbf{C}$ is constant. To be concrete, $\mathbf{C}$ is the Kronecker product of (1) the outer product of two all-one vectors of dimension $D$ and (2) an identity matrix of dimension $H \times H$. (Proposition 1)
>
> Then a natural next step is to make the core tensor trainable and inspect the changes in the expressive power (Theorem 3).
>
> The development of THSA does not rely on an exhaustive search of possible tensor structures but an observation of its tensorized representation. We call it Tuformer for two reasons. Firstly, it has a similar structure as Tucker Form (in traditional tucker form, a 4-way decomposition of 4 tensors while in THSA, the core tensor is a matrix). Secondly, we generalize the notion of heads to the stable rank of the core matrix $\mathbf{C}$, making the number of heads tunable.
>
> Besides introducing a design space, we hope to bring the tensor diagram to the ML community as an intuitive and strong tool for structural analysis from a tensor perspective. In addition, the Tuformer itself could advance the performance of many tasks across various domains with a guaranteed higher expressive power and a data-driven structure.
>
> We want to thank Reviewer R5aP for pointing out several issues in the paper writing, and we will make corresponding revisions and upload it soon.

---

> > ### Comment · Reviewer_R5aP · 2021-11-30
> > **Updated Paper**
> >
> > I looked again at the revised paper. Indeed, this is a curious case where lowering the aspirations and being less general makes the paper better: as the paper now no longer claims a relation to tensor decompositions but just puts forward an analysis of MHSA and proposes to add a learnable C, the story is coherent. Of course this step back also means that it is now a very incremental contribution, basically just proposing to learn rather than fix one additional parameter. But given that it takes some thinking to identify that "parameter", and that even simpler ideas like learning the leakage rate of the ReLU were worth paper, it might just be enough. I'd be fine to raise my recommendation from 6 to 7, although perhaps not quite to 8.

---

### Official Review · Reviewer_cD8e · 2021-11-01

**Correctness:** 2
**Technical Novelty And Significance:** 3
**Empirical Novelty And Significance:** 2
**Recommendation:** 6
**Confidence:** 4

**Main Review:**

Questions
===

How can a "super node" in Fig.4 be seen as a single tensor if a contraction of it crosses the boundary of a non-linear operation? This overloads the tensor diagram notation in an undesirable way, violating the multilinearity assumption. Taking this notation to the extreme, an MLP can be represented as a single "super-node" matrix, which spans multiple nonlinearities. Such a convention for tensor diagrams would be not much more useful than a typical block scheme of most neural architectures.

Tucker decomposition of a 4d-tensor leads to a 4d core tensor. If the core is a matrix, the authors should either carefully specify which "variant" of the decomposition is used (with specific references to external sources) or use a different term for the proposed parameterization.

The notation of T-junctions used to connect "stacked" modes to other nodes (such as to the matrix C from the top in Fig.4b) is crucial for understanding the proposed module; however, it is missing.

"The concept of the number of heads in THSA generalizes to the stable rank of the core matrix" -- why can't we draw a similar connection with the standard MHSA?

Related to the previous question, formulas (2) operate an integer number of heads (R) - how does this go along with the claim of a learned number of heads?

Formula (2b) rewrites a product of QK^T matrices for a single head through indexing along the R mode using s subscript. This corresponds to the rank factorization of the said product. This also seems to be equivalent to a single-head self-attention with R features instead of D, due to the fact that the values C_{rs} can be absorbed into either W^Q or W^K (which are in turn head-specific and may need an _{[r]} subscript). Or, if matrices W^Q and W^K are not head-specific anymore, this looks like a weight sharing scheme, in which one of the matrices is head-specific, and the other one is reused across heads. The main questions are: (1) where is Tucker decomposition in this formulation, (2) how does this equation correspond to Fig.4b, and (3) how exactly is this different from a standard MHSA?

Could the empirical advantage be due to a changed parameterization of W^K and W^Q matrices? I would like to see an experiment with the same configuration as MHSA in terms of dimensions of activations (probably corresponds to +25% configuration from the paper) but done in a way that the fifth matrix C is introduced into the QK product (using small surgical changes to the original MHSA module code). Such an added matrix experiment would level the changes with the proposed THSA and allow us to see whether the improvement is due to such parameterization of the QK product. Since such decomposition does not increase the expressive power of the QK product, any change in performance is due to either a different initialization or learning rate of the product elements resulting from such parameterization. If weight sharing is the case, additional studies of sharing W^Q or W^K in an MHSA setting are required.

The claim about the possibility of initializing THSA from MHSA should be followed with a remark that this is only possible in the case of matching dimensions.

The variance across runs in Fig 5 looks small - was RNG seed changed explicitly between runs to affect weight initializations, as opposed to relying on the intrinsic variance of SGD and randomness of multi-processing data loading in pytorch?

Around formulas (2), the first dimensions of Q and X^Q are F and N, respectively, but how is this possible if Q=X^Q W^Q?


Delivery
===
Abstract: "the number of heads is trainable" is misleading, as this number is discrete and is not directly optimized.

Abstract: "allows initialization from existing pre-trained models" is too broad, given that this is only possible in cases with a specific R.

The first occurrence of "expressive power" (except abstract) requires a clarification of the subject (a class of functions) - either as a citation or as a forward reference to the definition from the supplementary materials. It should also be made clear how expressive power is compared to explain when used together with "higher".

"(2) The number of heads is not trainable": not clear how this is a limitation or a problem.

"re-parameterization of the weight matrices via a variant of Tucker Decomposition or Tucker Representation" - the provided reference is too broad given the word "variant". A default Tucker decomposition assumes a core tensor of the same dimensionality as the decomposed tensor, which is not the case in this paper.

The proposed visualization of the softmax operation in the tensor diagram is good but requires clarification that normalization is applied along the mode containing the "alpha" point.

The "Evaluatuon order" section would benefit from a remark that the order affects the computational complexity of the overall contraction.

While the notation of stacking matrices to obtain a 3d-tensor is clear, connecting these stacked modes together requires additional explanation. Furthermore, if this notation is used purely to denote the stacked nature of nodes, it should be made clear, too (and perhaps such edges can be color-coded differently).

The related work section misses an overview of tensor decompositions used in machine learning.


Writing
===

Usage of the word "guaranteed" throughout the text requires explaining: by what means? If through a proof - the word "proven" is better suited.

"(2) The number of heads is not trainable ... capture different context information through each head": if this is a known fact, the authors should consider adding a reference where this is stated.

The first occurrence of "tensor representation theory" - a reference is required (e.g., Kolda)

"we also propose a novel data-driven structure that" - too vague.

"re-parameterization of the weight matrices" - which ones / of what?

"for instant, in Section 2.1" -> "for instance, in Fig. 2.3"

notation for subscript "_{[h]}" not introduced

Texts of "Single-head self-attention in tensor diagram" and "Multi-head self-attention in tensor diagram" sections are nearly identical; Fig. 3a is never referenced nor explained.


Code
===
The provided source code is not readily usable for reproducibility. A few problems and recommendations regarding the file with the implementation referred to in the readme:

1. The code contains multiple python syntax errors
2. Local variable tucker_heads is not defined in a few places; perhaps "self." is missing
3. Unused code clutter (all branches where self.tucker_heads is False)
4. Unused variables (e.g., c = self.c_proj(...))

**Summary Of The Paper:**

The paper investigates the multi-head self-attention mechanism (MHSA) of transformer networks through the lens of tensor decompositions via tensor diagram notation. The authors propose an extension to MHSA inspired by the Tucker decomposition (termed THSA), analyze its expressive power, and demonstrate that it belongs to a class of more expressive functions than MHSA. Further, the authors show the positive effect of this drop-in replacement on the downstream tasks.

**Summary Of The Review:**

The approach to analyzing MHSA taken by the authors is interesting and worth digging deeper, especially due to the clear empirical advantage of the proposed modification. However, the paper contains several issues in claims, theory, and connections between them. On the side of delivery, the paper has good potential, but in its current state, it is not well-polished and requires a major improvement to facilitate better understanding.

---

> ### Author Response · Authors · 2021-11-16
> **To Reviewer cD8e: Clarification of some wording related concerns and the model structure.**
>
> #### Responses on delivery
>
> > "The number of heads is trainable" is misleading, as this number is discrete and is not directly optimized.
>
> In MHSA, the number of heads H is discrete. In this paper, we generalize the number of heads to the **stable rank** of the core matrix ${\bf C}$, and it is continuous.
>
> In MHSA, the stable rank is the pre-set constant H as C is constant in MHSA --- it is the Kronecker product of **(a)** the outer product of two all-one vectors of dimension $D$ and **(b)** an identity matrix of dimension $H \times H$.
>
> In THSA, ${\bf C}$ is fully trainable. Thus its stable rank is trainable and data-driven. $R$ is the upper bound of its stable rank. That is why we refer to it as "trainable heads".
>
> > Abstract: "allows initialization from existing pre-trained models" is too broad, given that this is only possible in cases with a specific R.
>
> People use pre-trained models on downstream tasks to improve task performance and save computation and memory. In this paper, we show that Tuformer is compatible with any existing Transfomer-based pre-trained models and how to perform such initialization. Whenever using such models, the hyper-parameters(like $R$/$H$/$D$) should comply with the pre-trained model. The limitation is given in the original models, not Tuformers.
>
> ---
>
> We thank the reviewer for other suggestions and will make corresponding revisions. Please see previous comments for repeated explanations.
>
> We have added the requested experimental results. We have also clarified some misunderstandings.
>
> We also apologize for the typos and erros in version control of the code. We have uploaded the correct version.
>
> Please let us know if Reviewer cD8e has more questions! Thank you very much!

---

> ### Author Response · Authors · 2021-11-16
> **To Reviewer cD8e: Clarification of some wording related concerns and the model structure.**
>
> > Q5 The main questions are: (1) where is Tucker decomposition in this formulation, (2) how does this equation correspond to Fig.4b, and (3) how exactly is this different from a standard MHSA?
>
> (1) It is not a standard tucker decomposition where the core is 4-dimensional. Instead, the core tensor is 2-dimensional as denoted in Equation (2b). We refer to it as a variant of Tucker representation to spark future interests in exploration of Transformer parameterization from a tensor representation perspective. The proposed structure is actually inspired by the observation of tensor diagram representation of MHSA (See Proposition 1 and its proof in the appendix). We will rephrase the motivation in the revised version.
>
> (2) Equation (2b) is not equivalent to a single head attention with hidden dimension $R$. The main difference is that there is no inner product along with the hidden dimension $D$ compared to the single head attention. This difference is shown in Figure 4a and Figure 4b. Also, in Equation 2b, $Q_s$, $K_s$ are column vectors rather than matrices as in the single head self attention.
>
> (3) In MHSA, the core matrix $\mathbf{C}$ is constant while in THSA, the core matrix $\mathbf{C}$ is fully trainable.
>
> > Q6 Could the empirical advantage be due to a changed parameterization of W^K and W^Q matrices? I would like to see an experiment with the same configuration as MHSA in terms of dimensions of activations (probably corresponds to +25% configuration from the paper) but done in a way that the fifth matrix C is introduced into the QK product (using small surgical changes to the original MHSA module code). Such an added matrix experiment would level the changes with the proposed THSA and allow us to see whether the improvement is due to such parameterization of the QK product. Since such decomposition does not increase the expressive power of the QK product, any change in performance is due to either a different initialization or learning rate of the product elements resulting from such parameterization. If weight sharing is the case, additional studies of sharing W^Q or W^K in an MHSA setting are required.
>
> We are not sure if we understand this question correctly, but our interpretation is that the reviewer is concerned that the performance gain comes from **1)** the difference between the initialization of stacked and separate matrices and **2)** change of the normalization constant in $\mathsf{softmax}$.
>
> In our implementation, we follow the same initialization pattern as MHSA. As for the change of the normalization constant from $D$ to $R$ in $\mathsf{softmax}$ function, the purpose of the scaling is to make the distribution to approximate a 0-1 Gaussian distribution. Thus the change is required and does not affect the expressive power.
>
> The reviewer also mentioned an experiment in which MHSA and THSA have the same configuration.
> If the experiment the reviewer expects to see is a MHSA with a fixed core tensor of dimension $R \times R$, with 100% parameters in a single run, MHSA has BLEU score of 27.1 and THSA has BLEU score of 28.3.
>
>
>
> > Q7 The variance across runs in Fig 5 looks small - was RNG seed changed explicitly between runs to affect weight initializations, as opposed to relying on the intrinsic variance of SGD and randomness of multi-processing data loading in PyTorch?
>
> We did use different seeds in the experiments except for  Automatic Speech Recognition (ASR) in which we realized we accidentally reused the random seeds after double-checking the implementation. We are still re-running the ASR experiments and we do not see much variations so far.
>
>
> > Q8 Around formulas (2), the first dimensions of Q and X^Q are F and N, respectively, but how is this possible if Q=X^Q W^Q?
>
> The dimension of the matrix $\mathbf{Q}$ is $F \times R$ in the paper.
>
> > Q9 The claim about the possibility of initializing THSA from MHSA should be followed with a remark that this is only possible in the case of matching dimensions.
>
> We emphasize the fact that a THSA can be initialized from a pre-trained MHSA is exciting and makes THSA practical in real-world applications. The reviewer is right that only a THSA that matches the dimensions of MHSA could be initialized appropriately from a pre-trained MHSA. Although we believe it is implicitly self-explained, we agree that remarking it is useful.

---

> ### Author Response · Authors · 2021-11-16
> **To Reviewer cD8e: Clarification of some wording related concerns and the model structure.**
>
> We appreciate Reviewer cD8e's effort in reviewing this paper. We hope that our response could clarify some wording-related concerns and potential misunderstanding of the proposed Tuformers.
>
> #### Responses to the questions
>
> >Q1 How can a "super node" in Fig.4 be seen as a single tensor if a contraction of it crosses the boundary of a non-linear operation?
>
> We use "super node" to intuitively illustrate the potential design space among the weight matrices $\mathbf{W}^Q$, $\mathbf{W}^K$, $\mathbf{W}^V$, and $\mathbf{W}^O$. However, since there is a non-linear softmax operation, the freedom of design is limited. We will revise the corresponding wording and make it clearer to the readers.
>
> >Q2 Tucker decomposition of a 4d-tensor leads to a 4d core tensor. If the core is a matrix, the authors should either carefully specify which "variant" of the decomposition is used (with specific references to external sources) or use a different term for the proposed parameterization.
>
> The concerns in Q2 mostly come from our reference of the THSA as "a variant of Tuformer". The reason is that the proposed structure looks similar to a Tucker form. However, rather than a 4-dimensional tensor, the core tensor is 2-dimensional. It is inspired by the tensor diagram representation of MHSA (as detailed in Proposition 1 in Section 3). By studying the tensor diagram of MHSA, we found that MHSA is a special case of THSA with the core tensor ${\bf C}$ being a constant matrix (the multiplication of the outer product of two all-one vectors of dimension $D$ and an identity matrix of dimension $H \times H$). Then by making the core tensor to be trainable rather than fixed, we derived THSA.
>
> We refer to it as inspired by the Tucker tensor form because the proposed structure has a similar form as Tucker Tensor Decomposition. We thank Reviewer cD8e for pointing out that the word “inspired” may be misleading, and we will rephase it.
>
>
>
> > Q3 The notation of T-junctions used to connect "stacked" modes to other nodes (such as to the matrix C from the top in Fig.4b) is crucial for understanding the proposed module; however, it is missing.
>
> The T-junctions here consist of a batch multiplication and a contraction. We introduce these two operations and specify how to evaluate them (the evaluation can be done in any pair-wise sequence ordering, except for what is in the softmax box) in Section 2.1.
>
> > Q4 "The concept of the number of heads in THSA generalizes to the stable rank of the core matrix" -- why can't we draw a similar connection with the standard MHSA?"
>
> > Related to the previous question, formulas (2) operate an integer number of heads ($R$) - how does this go along with the claim of a learned number of heads?
>
> This generalization is consistent in MHSA and THSA. In MHSA, the stable rank is a constant because the core matrix $\mathbf{C}$ is the Kronecker product of (1) the outer product of two all-one vectors of dimension $D$ and (2) an identity matrix of dimension $H \times H$. Thus the stable rank of MHSA is $H$ and it will not change in training. In THSA, $\mathbf{C}$ is made trainable. The integer $R$ is the theoretical upper bound of the stable rank of $\mathbf{C}$.

---

> > ### Comment · Reviewer_cD8e · 2021-11-21
> > **Re: Tucker**
> >
> > The authors insist on calling the proposed attention mechanism Tucker, or "inspired by Tucker", despite the reviewers expressing confusion about this matter. The authors also admitted that the proposed (extended) tensor network has nothing to do with the Tucker decomposition, which according to the definition, assumes a 4D core tensor and matrices to increase mode dimensions. The authors also did not provide references to a source where such a decomposition is called Tucker. The authors also did not upload a revised PDF for the reviewers to see how they will handle this matter in the final version upon acceptance.
> >
> > The rationale "a Tucker decomposition, but with a matrix C instead of the core" is unconvincing and can be used more generally as "an X, but with Y", where Y does not fit the accepted definition of X. In the case of this paper, X is in the title, thus confusing the readers and making it harder to distantiate from the term.
> >
> > The machine learning community has adopted overloading notations and assigning different meanings to the existing terms. The usage of unrelated terms contributes to the lack of rigor and downweighs the otherwise good idea and experimental results. This lack of rigor makes the field cluttered and probes the boundaries of the scientific approach. The authors should consider fixing the source of confusion by renaming the method instead of patching the rationale.

---

> > > ### Author Response · Authors · 2021-11-22
> > > **Update of Revision**
> > >
> > > We thank Reviewer cD8e for the feedback.
> > >
> > > In the revised submission, we removed the association between Tuformer and Tucker Tensor Representation from the manuscript to avoid misunderstandings. We also re-organized the structure and added extensive details in the motivation of THSA (and Tuformer) and how THSA (and Tuformer) is derived.
> > >
> > > Please let us know if there are further questions or comments.

---

> > ### Comment · Reviewer_cD8e · 2021-11-29
> > **Feedback on the second revision**
> >
> > I have read the second revision, and for the most part it is a solid iteration on top of the original draft. The authors have found an elegant way to sidestep from the Tucker motive without changing the method name or title (except for just one remaining occurrence of "Tucker" somewhere in a figure caption). While the experiments are looking good, and the motivation is solid for the most part, I find many crucial comments raised in my original review and other reviewer's feedback addressed in an unsatisfying way. Particularly, I agree with Reviewer R5aP that the theoretical contribution is trivial. The second iteration also does not address generic out of proportion statements, such as the ability to initialize Tuformers from transformers, whereas this is only possible in a very specific case. Overall, I appreciate the effort with this submission, and for that I am increasing my score to "WA".

---

> > > ### Author Response · Authors · 2021-11-30
> > > **To reviewer cD8e: thank you for the feedback and clarification of the theoretical contributions.**
> > >
> > > We thank Reviewer cD8e for the feedback and are encouraged that the reviewer considered our paper well-motivated and sufficiently supported by our experiments. We hope we could also clarify some concerns in our theoretical contributions and the potential applications of Pre-trained models.
> > >
> > > When analyzing the expressive power of a neural network, we care about the approximation ability, whether the target function lies in the given functional space. A more expressive neural network means it has a higher probability to approximate the target function (which maps the input to the output) well. In this paper, we prove that Tuformers have higher expressive power than Transformers and this increase in the approximation ability is also reflected and supported by the experimental results in various domains.  As for the pre-trained model, we forget to emphasize that Tuformer can only be initialized in that specific way and we will definitely add that.
> > >
> > > We would really appreciate it if the reviewer could update the change of score in the main review as the end of the discussing period is approaching.

---

### Official Review · Reviewer_5BZt · 2021-11-01

**Correctness:** 2
**Technical Novelty And Significance:** 3
**Empirical Novelty And Significance:** 3
**Recommendation:** 5
**Confidence:** 4

**Main Review:**

Pros:
- Clear writing.
- The proposed idea is somewhat novel and interesting.

Cons:
- Motivation of THSA is not clear. Although the paper claims "inspired by a Tucker tensor form", what is the advantage of the Tucker tensor form? And why MHSA can be benefited from the Tucker tensor form?
- The ability of extended tensor diagram is limited. A traditional tensor diagram need not consider the combination order for diagram nodes. However, the proposed extension requires contracting the nodes in the softmax box firstly. I agree that the diagram extension is sufficient to present the proposed method. Nevertheless, it may need to elaborate the limitation clearly, for example,
	- Difficult to denote multi-layers due to the order of softmax box;
	- More attention when applying the tensor diagram extension to other cases besides the self-attention condition.
- More situations are supposed to be considered as the diagram extension is firstly proposed:
	- Is there a probability to take a node out from the box? If possible, what transformation needs to be applied?
	- Will there be a case multi-box overlap? When this case happened, how to do it?
- In Definition 2, the expressive power seems not to make sense. Under the definition, a model with higher expressive power may not perform well. For instance, an over-fitting model may have large expressive power, however, its performance is lower, compared to a suitable model. Therefore, it may not provide a guarantee to THSA performance. By the way, I guess there is a typo in Definition 2, "$f \in \mathcal{X}$" - > "$f \in F$".
- In Table 1, the paper seems not to mention that the results are all derived through re-implementing. And the results in the original paper are higher (e.g., ALBERT: MNLI-91.3%, QNLI-99.2%), which makes a concern that the experiment is not prepared well.
- Insufficient experiments. There are plenty of new works proposed recently. And if THSA is really consistently better than MHSA, THSA is supposed to achieve SOTA results on a task.
- More training run-time for THSA. Will THSA (same num of param) be slower when compared with MHSA?

**Summary Of The Paper:**

Focusing on the multi-head self-attention (MHSA) structure, this paper proposes an extension of the tensor diagram to denote self-attention (SA) structures more intuitively. Then inspired by the Tucker format, this paper also proposes a new form of SA named Tucker-Head Self-Attention (THSA), which can also be illustrated with the tensor diagram extension. Although THSA derives some success, THSA may lack theoretical guarantee and experiments seems insufficient.

**Summary Of The Review:**

Although this paper has a good presentation, for lacking theoretical guarantee and insufficient experiments, I regret to give a weak rejection.

---

> ### Author Response · Authors · 2021-11-16
> **To Reviewer 5BZt: Clarification of Motivation and Technical Details, Updated Experiment Results (II)**
>
> We appreciate Reviewer 5BZt’s effort in reviewing the paper. We will clarify the questions raised by the reviewer in the following discussion.
>
> >Q5 In Table 1, the paper seems not to mention that the results are all derived through re-implementing. And the results in the original paper are higher (e.g., ALBERT: MNLI-91.3%, QNLI-99.2%), which makes a concern that the experiment is not prepared well.
>
> The reason why the reported result is lower is that we did not use the largest possible ALBERT model and we did that using a single-model setup not an ensemble one. However, the comparison is still fair because they are based on the same pre-trained models (initializations), and the only difference is that the MHSA has a constant core matrix ${\bf C}$, and THSA has the trainable ${\bf C}$.
>
> We are repeating the experiment using the best model we can find online( huggingface ALBERT-xxlarge v2 model), the MNLI accuracy of which is 90.8%, and the QNLI is 95.3% using a single-model setup. The THSA achieves an accuracy of 91.7% in MNLI and 96.1% in QNLI. We will update other results in the revision once we have them.
>
>
> >Q6 Insufficient experiments. There are plenty of new works proposed recently. And if THSA is really consistently better than MHSA, THSA is supposed to achieve SOTA results on a task.
>
> We are wondering if Reviewer 5BZt could provide more information on which tasks are expected? We believe our experiments are extensive and sufficient since we (1) experiment on a suite of applications and datasets; (2) experiment on a wide range of model scales (model sizes) for each dataset; (3) experiment on both models trained from scratch and pre-trained large scale models.
>
> > Q7 More training run-time for THSA. Will THSA (same num of param) be slower when compared with MHSA?
>
> The result has been updated to the manuscript (Figure 6).
>
> Again, we thank Reviewer 5BZt for providing comments and feedback to our manuscript. We have added the requested experimental results. We have also clarified some misunderstandings. Please let us know if Reviewer 5BZt has more questions! Thank you very much again!

---

> > ### Comment · Reviewer_5BZt · 2021-11-30
> > **Thanks for the detailed response**
> >
> > Thanks for the detailed response. I would like to maintain my score.
> >
> > The proposed tensor diagram extension is interesting. Although it is helpful to present the Tuformer intuitively, it is not necessary for the proposed structure that can be analyzed from equation formulation. On THSA, on one hand, with a more generalized structure on MHSA, THSA has more expressive power. However, this expressive power may not promise performance in practice. And on the other hand, as mentioned by Reviewer cD8e and R5aP, the theoretical contribution is minor. Taking all into consideration, I may suggest more experiments on more MHSA based models for performance validation to increase persuasion.

---

> ### Author Response · Authors · 2021-11-16
> **To Reviewer 5BZt: Clarification of Motivation and Technical Details, Updated Experiment Results**
>
> We appreciate Reviewer 5BZt’s effort in reviewing the paper. We will clarify the questions raised by the reviewer in the following discussion.
>
> > Q1 Motivation of THSA is not clear. Although the paper claims "inspired by a Tucker tensor form", what is the advantage of the Tucker tensor form? And why MHSA can be benefited from the Tucker tensor form?
>
> As mentioned in the introduction section, the primary motivation of Tuformer Models and THSA is three-fold: **(1)** provide theoretical support and potential guidance to the structural design of MHSA by analyzing the expressive power of the network with the help of the tensor diagram representation, **(2)** make the number of heads trainable, and **(3)** provide a design space for future enhancement of model performance.
>
>
> By studying the tensor diagram of MHSA, we found that MHSA is a special case of THSA with the core tensor ${\bf C}$ being a constant matrix (the multiplication of the outer product of two all-one vectors of dimension $D$ and an identity matrix of dimension $H \times H$). Then by making the core tensor to be trainable rather than fixed, we derived THSA.
>
> We refer to it as inspired by the Tucker tensor form because the proposed structure has a similar form as Tucker Tensor Decomposition. We thank Reviewer 5BZt for pointing out that the word “inspired” may be misleading, and we will rephase it.
>
> > Q2 The ability of extended tensor diagram is limited. A traditional tensor diagram need not consider the combination order for diagram nodes. However, the proposed extension requires contracting the nodes in the softmax box firstly. I agree that the diagram extension is sufficient to present the proposed method. Nevertheless, it may need to elaborate the limitation clearly, for example,
> > - Difficult to denote multi-layers due to the order of softmax box;
> > - More attention when applying the tensor diagram extension to other cases besides the self-attention condition.
>
>
> The tensor-diagram itself is nothing but a precise graphical representation of the math equations.
>
> The $\mathsf{softmax}$ box is an element-wise operation and will not affect the representation of multi-layer structures. Since we focus on the analysis of the expressive power and improvement of MHSA, the discussion of other extensions of the tensor diagram is beyond the scope of this paper.
>
>
> Moreover, whether there is a probability to take a node outside the softmax box and what transformation needs to be applied is equivalent to the question that: Given $\mathsf{softmax}(AB)$ where $A$ $B$ are matrices, whether we can take $A$ or $B$ outside the $\mathsf{softmax}$ function. This is a math question rather than the limitation of the tensor diagram.
>
>
> >Q3 More situations are supposed to be considered as the diagram extension is firstly proposed:
> Is there a probability to take a node out from the box?
> > - If possible, what transformation needs to be applied?
> > - Will there be a case multi-box overlap? When this case happened, how to do it?
>
> We are not sure whether we understand the reviewer correctly. If the question is how to deal with the overlapped $\mathsf{softmax}$ boxes, there are two possible cases: **(1)** one box contains another box, **(2)** one box partially overlaps with another one.
>
> This is still a math question. In case **(1)** (one box contains another box), it is equivalent to $\mathsf{softmax}(\mathsf{softmax}({\bf A}))$, we first evaluate the inner $\mathsf{softmax}$ then the outer one. Case **(2)** is not allowed because something like $\mathsf{softmax}$ like $\mathsf{softmax}({\bf A} \mathsf{softmax}[{\bf B}) {\bf C}])$ is invalid in math.
>
>
> > Q4 In Definition 2, the expressive power seems not to make sense. Under the definition, a model with higher expressive power may not perform well. For instance, an over-fitting model may have large expressive power, however, its performance is lower, compared to a suitable model. Therefore, it may not provide a guarantee to THSA performance. By the way, I guess there is a typo in Definition 2, " " - > " ".
>
> We first thank Reviewer 5BZt for pointing out the typo. As for the definition of expressive power, we politely point out that the approximation ability and the generalization ability are different perspectives of a neural network. When analyzing the expressive power of a neural network, we care about the approximation ability, whether the target function lies in the given functional space. A more expressive neural network (whose parameters need to be trained and thus is a function class) means it has a higher probability to approximate the target function (which maps the input to the output) well. The overfitting issue, however, means that the training data is not very representative of the testing data, which is a generalization problem.

---

### Author Response · Authors · 2021-11-11
**Update of the supplementary material**

Dear Reviewers,
  The previous code was not correct due to an issue in version control.
  We apologize for the inconvenience and have updated it.
Best regards,
The Authors.

---

### Author Response · Authors · 2021-11-22
**Summary of Revision**

We thank all reviewers for their insightful questions and valuable feedback. We are particularly encouraged that they consider the tackled problem both interesting and important.
Here we briefly outline the updates to the revised submission based on the reviews. We address individual questions of reviewers in separate responses.

$\newcommand{Ro}{\textcolor{purple}{5BZt}}$
$\newcommand{Rt}{\textcolor{red}{cD8e}}$
$\newcommand{Rth}{\textcolor{blue}{R5aP}}$
$\newcommand{Rf}{\textcolor{green}{hfb5}}$

Paper Updates:

[General]  We completely removed the association between Tuformer and Tucker Tensor Representation from the manuscript to avoid misunderstandings. We also re-organized the structure and added extensive details in the motivation of THSA (and Tuformer) and how THSA (and Tuformer) is derived. ($\Ro$, $\Rt$, $\Rth$, $\Rf$). Then we changed the title accordingly.

[Section 3 Tuformer]  We revised the wording to avoid misunderstanding of the proposed design space.($\Ro$,  $\Rt$,  $\Rth$).

[Section 4 Experiment]  We added the inference run-time comparison($\Rf$), we updated the ALBERT experiments using the largest possible pre-trained models ($\Ro$ ).

[Section 5 Related Work] We added a review of tensor decomposition in machine learning.

---

### Decision · Program_Chairs · 2022-01-20

**Decision:**

Accept (Poster)

**Comment:**

This paper presents a tensor diagram view of the multi-headed self-attention (MHSA) mechanism used in Transformer architectures, and by modifying the tensor diagram, introduces a strict generalization of MHSA called the Tucker-head self attention (THSA) mechanism. While there is some concern regarding the incremental nature of the proposition, the identification of where to usefully add the additional parameter that converts from MHSA to THSA was nontrivial, and the experimental results on the performance benefits across multiple tasks is convincing.